# CBX3 antagonizes IFNγ/STAT1/PD-L1 axis to modulate colon inflammation and CRC chemosensitivity

Yao Xiang[1], Jorge Mata-Garrido [1], Yuanji Fu [1], Christophe Desterke [2], Eric Batsché [3], Ahmed Hamaï[1], Christine Sedlik[4], Youssouf Sereme[1], David Skurnik[1,5], Abdelali Jalil [6], Rachel Onifarasoaniaina [7], Eric Frapy[1], Jean-Christophe Beche[8], Razack Alao[8], Eliane Piaggio[4], Laurence Arbibe[1,9] & Yunhua Chang [1,9]✉

## Abstract

As an important immune stimulator and modulator, IFNγ is crucial for gut homeostasis and its dysregulation links to diverse colon pathologies, such as colitis and colorectal cancer (CRC). Here, we demonstrated that the epigenetic regulator, CBX3 (also known as HP1γ) antagonizes IFNγ signaling in the colon epithelium by transcriptionally repressing two critical IFNγ-responsive genes: *STAT1* and *CD274* (encoding Programmed death-ligand 1, PD-L1). Accordingly, CBX3 deletion resulted in chronic mouse colon inflammation, accompanied by upregulated *STAT1* and *CD274* expressions. Chromatin immunoprecipitation indicated that CBX3 tethers to *STAT1* and *CD274* promoters to inhibit their expression. Reversely, IFNγ significantly reduces CBX3 binding to these promoters and primes gene expression. This antagonist effect between CBX3 and IFNγ on STAT1/PD-L1 expression was also observed in CRC. Strikingly, CBX3 deletion heightened CRC cells sensitivity to IFNγ, which ultimately enhanced their chemosensitivity under IFNγ stimulation in vitro with CRC cells and in vivo with a syngeneic mouse tumor model. Overall, this work reveals that by negatively tuning IFNγ-stimulated immune genes' transcription, CBX3 participates in modulating colon inflammatory response and CRC chemo-resistance.

**Keywords** CBX3 (HP1γ); IFNγ Signaling; Colon; Inflammatory/Immune Response; Colorectal Cancer
**Subject Categories** Cancer; Digestive System; Immunology

## Introduction

Interferon γ (IFNγ or IFNG) is an essential cytokine in orchestrating both innate and adaptive immune responses. IFNγ signaling is activated by binding of IFNγ to its receptors (IFNγR1 and IFNγR2). This binding forms the IFNGR protein complex and subsequently activates JAK1/JAK2, which further phosphorylates and dimerizes the transcription factor STAT1 (signal transducer and activator of transcription). The phospho-STAT1 dimer then migrates to the nucleus and activates the inflammation response by promoting interferon-stimulated genes (ISGs) transcription, including STAT1 itself and IRF1 (interferon regulatory factor 1). The latter, in turn promotes CD274 (Programmed cell death ligand 1, PD-L1) transcription to downregulate the magnitude of the inflammation response. Therefore, as an important immune stimulator and modulator, IFNγ, along with its effectors STAT1 and PD-L1, plays very crucial roles in balancing immune homeostasis and inflammatory response. The aberrant IFNγ signaling is often associated with aberrant immunity provoking autoinflammatory/autoimmune diseases (such as ulcerative colitis) as well as oncogenesis (such as colorectal cancer; Kak et al, 2018; Yi et al, 2018; Du et al, 2022; Alspach et al, 2019).

The heterochromatin protein 1 (HP1) family are epigenetic regulators, which are readers of the H3K9me2/3 histone modifications and play an important role in the formation and maintenance of heterochromatin. HP1 family includes CBX1 (HP1β), CBX5 (HP1α), and CBX3 (HP1γ) in mouse and human. In this study, we demonstrated that an epigenetic regulator CBX3 (Chromobox protein homolog 3), also called HP1γ (Heterochromatin protein 1γ), antagonizes IFNγ signaling via directly repressing the transcription of two key interferon-stimulated immune genes, *STAT1* and *PD-L1*. This important IFNγ antagonist role placed CBX3 in a key position to keep colon immune homeostasis.

[1]Université Paris Cité, INSERM, CNRS, Institut Necker Enfants Malades, F-75015 Paris, France. [2]Université Paris-Saclay, INSERM, Laboratory of Modèles de cellules souches malignes et thérapeutiques, Villejuif F-94805, France. [3]Sorbonne Université, Institut de Biologie Paris-Seine, CNRS UMR8256 Biological Adaptation and Aging (IBPS), Laboratory of Epigenetics and RNA Metabolism in Human Diseases, 75005 Paris, France. [4]Institut Curie, PSL University, Department of Translational Research, Inserm U932, Laboratory of Immunity and Cancer, F-75005 Paris, France. [5]Service de Bactériologie, virologie, parasitologie et hygiène, AP-HP, Hôpital Necker, F-75015 Paris, France. [6]Université Paris Cité, CNRS, SPPIN - Saints-Pères Paris Institute for the Neurosciences, F-75006 Paris, France. [7]Université de Paris Cité, INSERM, CNRS, Institut Cochin, F-75014 Paris, France. [8]Laboratory of Expérimentation Animale et Transgénèse SFR Necker-Inserm US 24, Paris, France. [9]These authors contributed equally: Laurence Arbibe, Yunhua Chang. ✉E-mail: Yunhua.chang-marchand@inserm.fr

Ulcerative colitis (UC) is characterized by chronic inflammation of the colon and rectum (Danese and Fiocchi, 2011; Gajendran et al, 2019). The exact etiology of UC is not fully elucidated, but the dysregulated inflammatory response and immune system dysfunction are considered as the two most probable causes of UC. The dysregulated IFNγ signaling is often found in the pathophysiology of inflammatory bowel diseases (IBD) (Peterson and Artis, 2014; Neurath, 2014; Nava et al, 2010). High expression levels of IFNγ and PD-L1 were detected in the colon mucosa of UC and UC-associated dysplasia/colonic cancer (Beswick et al, 2018; Iacomino et al, 2020; Ozawa et al, 2021). Accordingly, an increased STAT1 expression and activation in the mucosa of IBD patients, particularly in UC, has also been documented (Schreiber et al, 2002).

Our recent work has revealed a significant decrease of CBX3 (HP1γ) expression in the colon epithelium of UC patients, and that loss of CBX3 leads to a decreased RNA splicing precision in UC (Mata-Garrido et al, 2022). Our previous work also showed that CBX3 participated in fine-tuning immune gene expression in response to enterobacteria infection (Harouz et al, 2014). Here, we identified that CBX3 works as an antagonist against IFNγ signaling, and its deletion induced a long-lasting inflammation in the colon epithelium. These studies have started to reveal a new role of CBX3 in controlling the colon epithelium inflammatory response, in addition to its classical role in the formation and stabilization of heterochromatin.

Colorectal cancer (CRC) is another frequent colon pathology and IFNγ signaling plays crucial roles in its development and treatment. These roles could be both immunostimulatory and immunosuppressive. All the factors determining these paradoxical roles of IFNγ in cancer are not yet completely clear. Much work is still needed to understand how to modulate IFNγ signaling to improve cancer therapy efficacy (Du et al, 2022; Alspach et al, 2019). Chemotherapy is the most used treatment strategy in colon cancer. IFNγ influence chemotherapeutic efficacy by regulating the immune effects of anti-cancer therapies as well as the ability of cancer cells to respond to genotoxic damage (Minn, 2015; Coffelt and de Visser, 2015). Here, we identified a new role of CBX3 as an antagonist of the IFNγ pathway in colon epithelium, which designated CBX3 as a potential target to enhance IFNγ related therapeutic effectiveness. Our further clinical analysis revealed also the antagonist effect between IFNγ and CBX3 in colorectal cancer on *STAT1/PD-L1* expression. Particularly, low CBX3 expression associated with better CRC patients' overall survival. Corresponding to this result, CBX3 depletion makes IFNγ-insensitive CRC cells dramatically regain IFNγ sensitivity, which significantly increases CRC cells' chemosensitivity under IFNγ stimulation.

Altogether, this work identifies a new antagonist interplay between CBX3 and IFNγ signaling to regulate colon immune response and chemosensitivity of colorectal cancer, which highlights CBX3 as a potential target to improve the treatment of UC and colorectal cancer.

# Results

## Conditional knockout of *Cbx3* in the mice gut epithelium induces a chronic inflammatory state in the colon

Different gut inflammation criteria were assessed in Villin-creERT2:*Cbx3*$^{-/-}$ mice (assigned as *Cbx3* KO mice in later text) at

7 days and 12 months post-tamoxifen induction (Fig. 1A,B) (Ulrike et al, 2014). Firstly, we checked immune cells infiltration in colon epithelium. CD4$^+$ T cells, Ly6G$^+$ neutrophil cells, and CD68$^+$ macrophages were all significantly increased in the proximal and distal colon of the *Cbx3* KO mice at day 7 post-tamoxifen induction. CD8$^+$ T cells significantly increased in the proximal but not in the distal colon of the *Cbx3* KO mice at day 7 post-tamoxifen induction (Fig. 1C–F).

Remarkably, infiltration of CD4$^+$ T cells, CD68$^+$ macrophage and Ly6G$^+$ neutrophil cells in the proximal colon was long-lasting, as shown by the increased detection in the *Cbx3* KO mucosa at 12 months post-tamoxifen induction (Fig. 1C–F). Furthermore, the size of lymphatic nodules became larger in *Cbx3* KO mice as compared to WT mice 12 months post-tamoxifen induction (Fig. 1G,H).

Moreover, Periodic Acid Schiff (PAS) staining revealed a clear reduction of goblet cells in both 7 days and 12 months *Cbx3 KO* colon (Fig. 2A). At the same time, the crypt length was significantly increased in *Cbx3 KO* colon (Fig. 2B). Likewise, the Ki67 and BrdU positive cells are also increased in 7 days *Cbx3* KO colon (Fig. 2C–E), which suggested an increased cell cycling in the *Cbx3*KO colon crypt. GSEA analysis based on RNA seq data from day 7 post-tamoxifen induction colon epithelium reflected the enrichment of G2/M checkpoint, mitotic spindle as well as E2F targets in *Cbx3* KO mice (Fig. 2F). Together, all the data indicate increased cell proliferation in colon crypt and epithelial hyperplasia.

Overall, these data indicate that CBX3 deficiency triggers long-lasting colon mucosal inflammation, demonstrated by goblet cell reduction, crypt length alterations, epithelial hyperplasia and increased immune cells infiltration.

## CBX3 targets *Stat1* and *Cd274* genes to repress their expression in vivo in the mice colon epithelium

Integrative analysis was applied to identify the direct gene target(s) of CBX3 (HP1γ) in the colon epithelium. We crossed the RNA-seq data obtained from purified *Cbx3* KO mice colon epithelium (GSE192800) (Data ref: Mata-Garrido et al, 2022), with a publicly available CBX3 Chromatin immunoprecipitation-sequencing (ChIP-seq) performed in HCT116, a frequently used colorectal cancer cell line (GSE28115) (Data ref: Smallwood et al, 2012).

Over 1100 genes' expression exhibited an over 1.5 times modification (increase or decrease) in RNA sequencing in comparing the colon epithelium of *Cbx3*KO mice to which of WT mice. 4942 genes revealed by ChIP-seq whose proximal promoters' regions (from 3000 bp upstream to 500 bp downstream of their Transcription Starting Site) possess at least a CBX3 binding site. By crossing these two databases, 68 genes were sorted as potential CBX3 target genes (Fig. 3A). Interestingly, among the upregulated genes, the STRING network analysis revealed a cluster composed of 10 CBX3 direct targets implicated in the immune response. Among them, *Stat1* and *Cd274*, two important components of the IFNγ signaling pathway (Fig. 3B), were recognized as CBX3 direct targets in colon epithelium. On these two genes, ChIP-seq evidenced the peaks of CBX3 associations at their promoter regions (Fig. 3C,D) and RNA-seq revealed that their transcription was significantly upregulated upon *Cbx3* inactivation (Fig. 3E,F). In parallel, GSEA analysis revealed significant enrichments of genes involved in the inflammatory and IFNγ responses in the *Cbx3* KO mice colon epithelium (Fig. 3G).

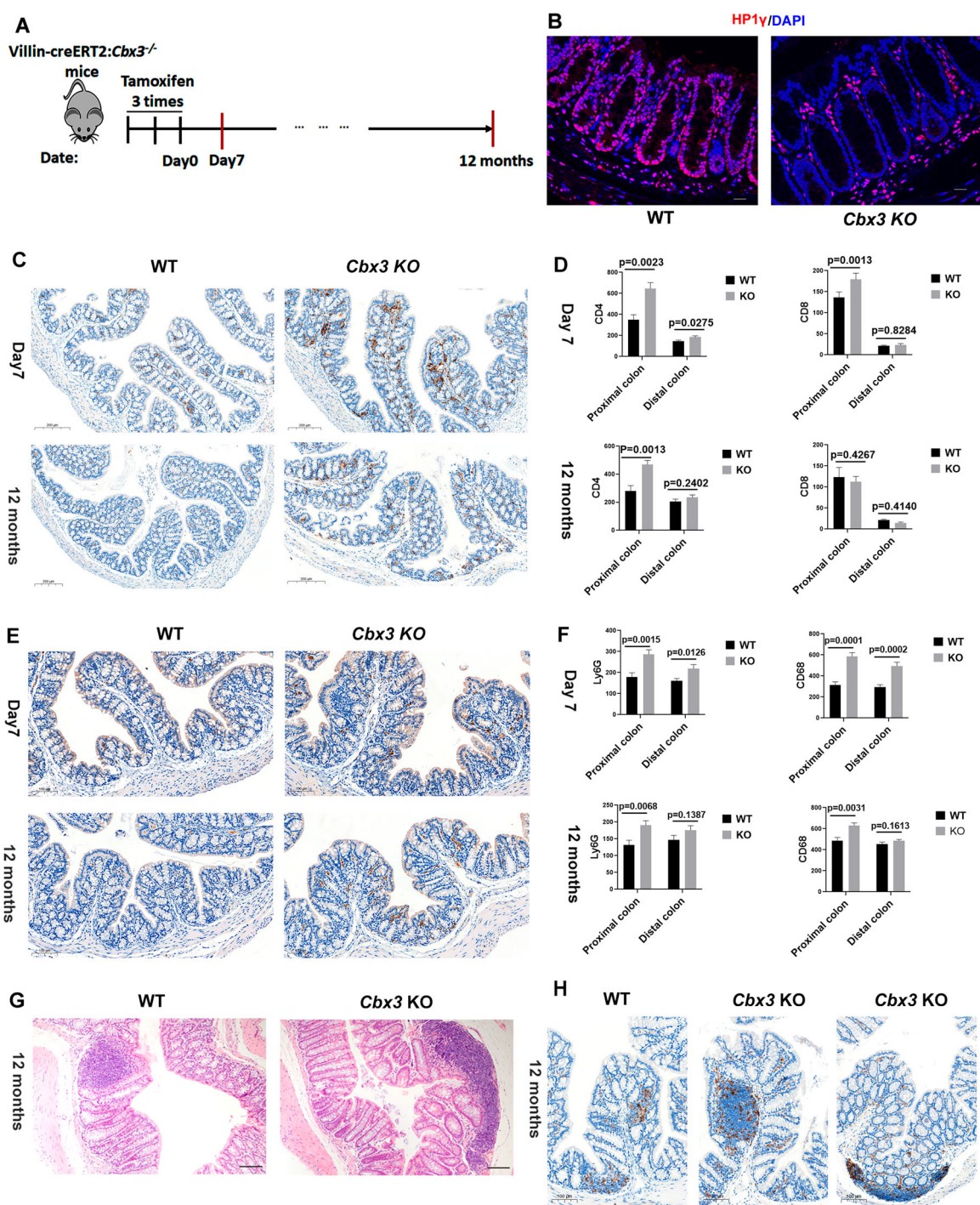

The *Stat1* and *Cd274* that appeared from bioinformatic analysis could be just considered as potential targets of CBX3, further analysis is required to validate the bioinformatics analysis. Firstly, the expressions of STAT1 and PD-L1 were examined with colon epithelium from *Cbx3* WT and *Cbx3* KO mice at 7 days and 12 months post-tamoxifen induction. The increased STAT1 and PD-L1 protein levels were detected in *Cbx3* KO colon epithelium (Fig. 4A,B). In parallel, RT-qPCR showed that deletion of *Cbx3* durably increased *Stat1* and *Cd274* mRNA levels in the colon epithelium even after 12 months of tamoxifen induction (Fig. 4C).

Likewise, immunofluorescence studies revealed a more intense STAT1 nucleus labeling in *Cbx3* KO colon epithelium (Fig. 4D). Overall, these results spotlight in vivo, *Cbx3* deletion increased STAT1 and PD-L1 expressions in the colon epithelium.

## CBX3 deficiency upregulates STAT1 and PD-L1 ex vivo expressions in colon organoid

In vivo results revealed that both STAT1 and PD-L1 expressions were increased in *Cbx3* KO colon epithelium. However, these

◀ **Figure 1. Conditional knockout of *Cbx3* in the mice gut epithelium induces a chronic inflammatory state in the colon.**

(A) The scheme illustrates the time-course of sample collection. Villin-creERT2:*Cbx3*[-/-] mice were assigned as *Cbx3* KO mice in the later text. The samples were collected at 7 days or at 12 months after three times Tamoxifen gavage. (B) Representative immunofluorescence in colon tissue sections from *Cbx3* KO and WT mice stained with anti-HP1γ (CBX3) antibody (red) and DAPI (blue). Scale bar: 20 μm. (C) Representative immunochemistry with anti-CD4 antibody in WT and *Cbx3* KO colon tissues. Upper panel: 7 days WT (left) and *Cbx3* KO (right) colon; Down panel: 12 months WT (left) and *Cbx3* KO (right) colon. Scale bar: 200 μm. (D) Cell counting revealed CD4[+] or CD8[+] cells infiltration in *Cbx3* KO colon 7 days or 12 months post-tamoxifen gavage. The Y axis represents CD4[+] or CD8[+] cell number counted in each 4 μm thickness colon section from proximal or distal colon (10–20 sections from 4 mice were counted of each distal or proximal group, two-sided Student's t-test). The error bar represented SEM (standard error of mean). Upper panel: 7 days CD4[+] (left) or CD8[+] (right) cell number counted in WT and *Cbx3* KO colon; Down panel: 12 months CD4[+] (left) or CD8[+] (right) cell number counted in WT and *Cbx3* KO colon. (E) Representative immunochemistry with anti-Ly6G antibody in WT and Cbx3 KO colon tissues. Hematoxylin (blue) counterstaining was used for nucleus coloration. Upper panel: 7 days WT (left) and *Cbx3* KO (right) colon; Down panel: 12 months WT (left) and *Cbx3* KO (right) colon. (F) Cell counting revealed increased Ly6G[+] or CD68[+] cells infiltration in Cbx3 KO colon 7 days or 12 months post-tamoxifen gavage. The Y axis represents Ly6G[+] or CD68[+] cell number counted in each 4 μm thickness colon section from proximal or distal colon (10–20 sections from 4 mice were counted of each distal or proximal group, two-sided Student's t-test). Scale bar: 100 μm Upper panel: 7 days Ly6G[+] (left) or CD68[+] (right) cell number counted in WT and *Cbx3* KO colon; Down panel: 12 months Ly6G[+] (left) or CD68[+] (right) cell number counted in WT and *Cbx3* KO colon. The error bar represented SEM. (G, H) Representative HE (Hematoxylin and eosin) staining and immunochemistry with anti-CD4 antibody revealed increased nodule lymphoid size in *Cbx3* KO mice colon 12 months post-tamoxifen gavage. Scale bar: 100 μm. Source data are available online for this figure.

transcriptional inductions could be due to higher IFNγ production in *Cbx3* KO colon resulting from increased T cell infiltration (Fig. 1C,D). To circumvent this possibility and establish a direct link between *Cbx3* KO and increased STAT1 and PD-L1 expressions, we generated colon organoids derived from WT or *Cbx3* KO mice and stimulated them in vitro with mouse IFNγ (Fig. 4E). Importantly, deletion of *Cbx3* resulted in enhanced expression of *Stat1* and *Cd274*, both under basal culture state and after 24 h IFNγ stimulation (Fig. 4F). CBX3 deficiency thus increased *Stat1* and *Cd274* expression under the same IFNγ concentration, suggesting CBX3 depletion is a direct and sufficient condition for enhancing *Stat1* and *Cd274* expression.

## CBX3 deficiency dramatically increases STAT1 and PD-L1 in vitro expression upon IFNγ stimulation

We then tried to study in vitro the interaction between CBX3 and IFNγ on STAT1/ PD-L1 expression by generating CRISPR/Cas9 CBX3 KO cells with SW480 and HT29, two commonly used CRC cell lines. The editing efficiency of the CRISPR/Cas9 was verified by RT-qPCR and Western blot analyses (Fig. 5A,B). A compensatory mechanism revealed by an increased mRNA expression of *CBX5* and *CBX1* was found in SW480 but not in HT29 CRISPR/Cas9 *CBX3* KO cells (Fig. EV1).

Remarkably, IFNγ stimulation limitedly increased STAT1 expression in WT cells, but strongly increased its mRNA and protein expression levels in two different *CBX3* KO cell lines (Fig. 5C,D). Western blot analysis using a phospho-STAT1 antibody further evidenced the increased active STAT1 specifically in *CBX3* KO cells under IFNγ stimulation (Fig. 5C,D, middle panels). Finally, immunofluorescence analysis visually confirmed the dramatically increased STAT1 expression in *CBX3* KO cells upon IFNγ stimulation (Fig. 5C,D, right panels). Moreover, we further applied cytometry analysis by using an anti-STAT1 Phospho (Tyr701) PE antibody to confirm the increased p-STAT1 level in *CBX3* KO cells. As shown, the IFNγ stimulation increased significantly p-STAT1 level in two different types *CBX3*KO cells compared to their WT control, but this increase is more evident in HT29 *CBX3* KO cells than in SW480 *CBX3* KO cells (Fig. 5E).

PD-L1 expression followed a similar trend. Upon IFNγ stimulation, both mRNA and cell surface protein level of PD-L1 much more increased in the two *CBX3* KO cell lines compared to WT control, as shown by RT-qPCR (Fig. 6A) and flow cytometry analyses (Fig. 6B,C).

Finally, knock down *STAT1* by shRNA in HT29 cells significantly deceased *CD274* expression in basal condition or under IFNγ stimulation, which confirmed the existence of IFNγ/STAT1/PD-L1 axis in CRC cells (Fig. 6D).

Hence, similar to the phenotype observed in the ex vivo model of colon organoid, CBX3 deficiency is sufficient for priming STAT1 and PD-L1 expressions in the CRC cells. Particularly, contrary to colon organoids which answer well to IFNγ, the CRC cells (HT29 and SW480) limitedly respond to IFNγ stimulus. However, CBX3 deficiency ultimately made these IFNγ-insensitive CRC cells becoming extremely sensitized to IFNγ stimulation.

## IFNγ stimulation reduces the binding of CBX3 to the promoters of *STAT1* and *CD274*

We next investigated the association of CBX3 to its target genes *STAT1* and *CD274*. CBX3 ChIP-Seq data revealed three CBX3 association peaks in *STAT1* and *CD274* genes (peaks 51235, 51236 for *STAT1* and peak 91126 for *CD274*) which should be localized at their promoter regions. Accordingly, the UCSC Genome Browser confirms that these CBX3 binding sites (the blue regions in Fig. 7A,D) are located exactly in the promoter regions of *STAT1* and *CD274* genes, which are enriched with H3K4Me3, H3K27Ac, CpG island regions, located just upstream of their transcription start codon (Figs. 7A,D and EV2).

ChIP assays for CBX3 were performed in the HT29 cell line with and without IFNγ stimulation. CBX3 chromatin association was detected by Q-PCR using primers specific to the identified binding sites, as peaks 51235, 51236 for *STAT1* and peak 91126 for *CD274* (Fig. 7A,D). Q-PCR with primer sets aside from indicated CBX3 binding domain (non-specific primers) are shown in Fig. EV3A.

In unstimulated HT29 cells, ChIP assays revealed a potent association of CBX3 to promoter regions for both genes. Interestingly, IFNγ stimulation significantly decreased CBX3 chromatin association on these sites (Fig. 7B,C,E). This decrease could not be attributed to a decrease of CBX3 level since 24 h IFNγ stimulation did not result in any significant change in the expression of CBX3 at both the mRNA (Fig. EV3B) and protein (Fig. 5C) levels in HT29 cells.

We also noted an uncharacterized long non-coding RNA (LOC105373805), which located upstream to the *STAT1* promoter in opposite orientation, most likely under the control of a

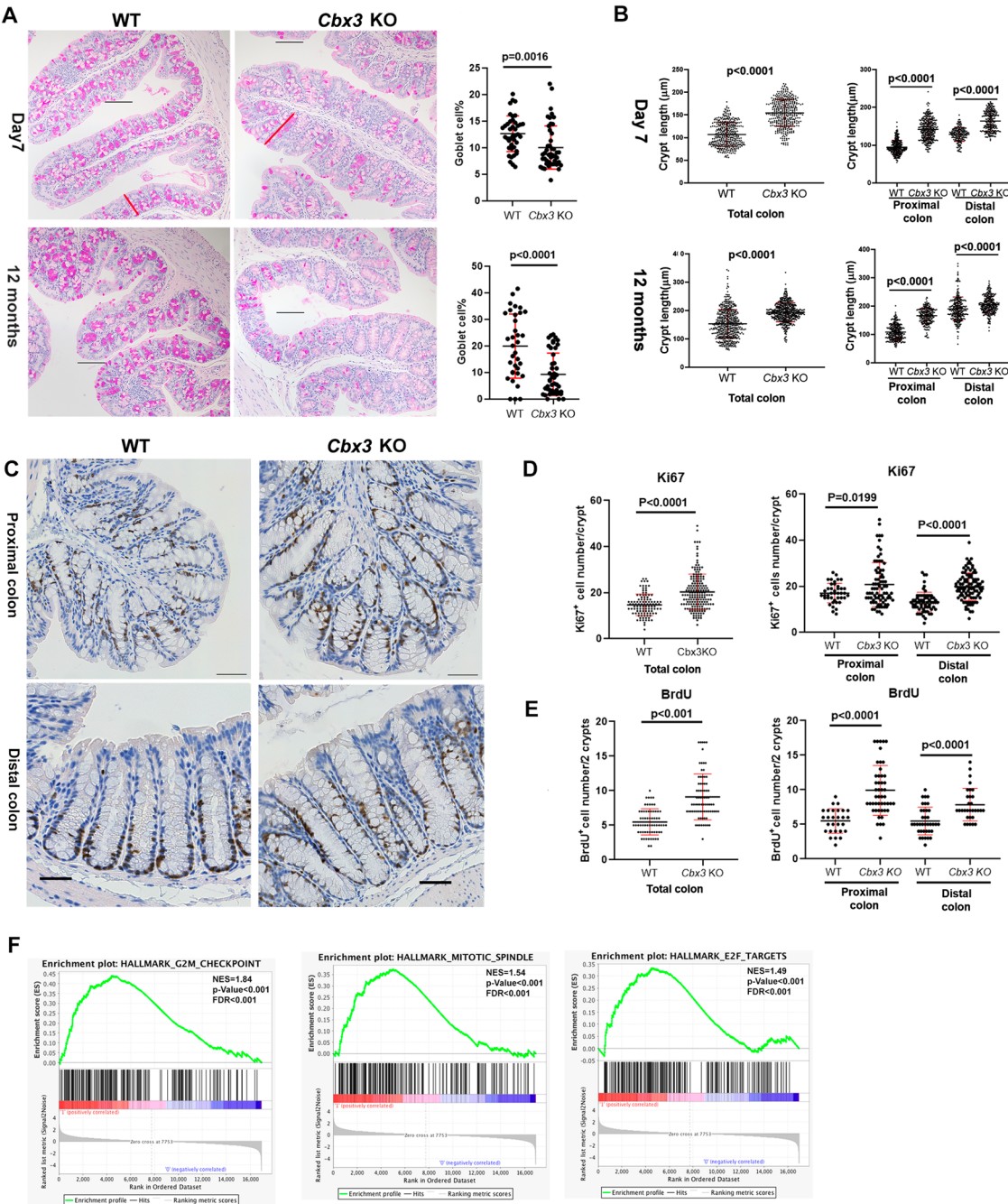

**Figure 2. Conditional knockout of *Cbx3* in the mice gut epithelium induces a chronic inflammatory state in the colon.**

(A) Representative sections of PAS (Periodic Acid Schiff) staining (left) and counting (right) revealed reduced goblet cells in *Cbx3* KO colon at the indicated times. The quantification of goblet cell is processed by using Ilastik (Berg, S. et al, 2019). The ratio between the area occupied by goblet cells and the area of the entire tissue was calculated by ImageJ software. Scale bar: 100 µm. The red trait indicated the measurement of a crypt with integrity structural (each group includes 16–32 sections from 4 mice, each section is presented in 1–3 images, totally 36–48 images were analyzed. Two-sided Student's t test). The data are shown as means ± SD (standard deviation). (B) Colon crypt length measured by Image J revealed a significant increase in *Cbx3* KO mice compared to WT (10–20 sections from 4 mice were measured of each distal or proximal group, two-sided Student's t test). All data are shown as means ± SD. (C) Representative immunochemistry with anti-Ki67 antibody in WT and Cbx3 KO mice colon tissues. Hematoxylin (blue) counterstaining was used for nucleus coloration. Scale bar: 50 µm. (D) 10–20 Crypts with integrity structural of each mouse has been counted (4 mice each group, two-sided Student's t test). The Y axis represents Ki67⁺ cell number counted in each crypt from proximal or distal colon, all data are shown as means ± SD. (E) Cell counting revealed increased BrdU⁺ in *Cbx3* KO mice colon 7 days post-tamoxifen gavage (5 sections/mice, 6–8 crypts/section). The Y axis represents Brdu⁺ cell number counted per 2 crypts from proximal or distal colon (4 mice in each group, two-sided Student's t-test), all data are shown as means ± SD. (F) GSEA analysis based on transcriptome data revealed a significant enrichment hallmark of G2M checkpoint, Mitotic spindle and E2F targets in Cbx3 KO mice colon epithelium. Signal2Noise uses the difference of means scaled by the SD with FDR (false discovery rate) adjustment for multiple testing, n = 3. Source data are available online for this figure.

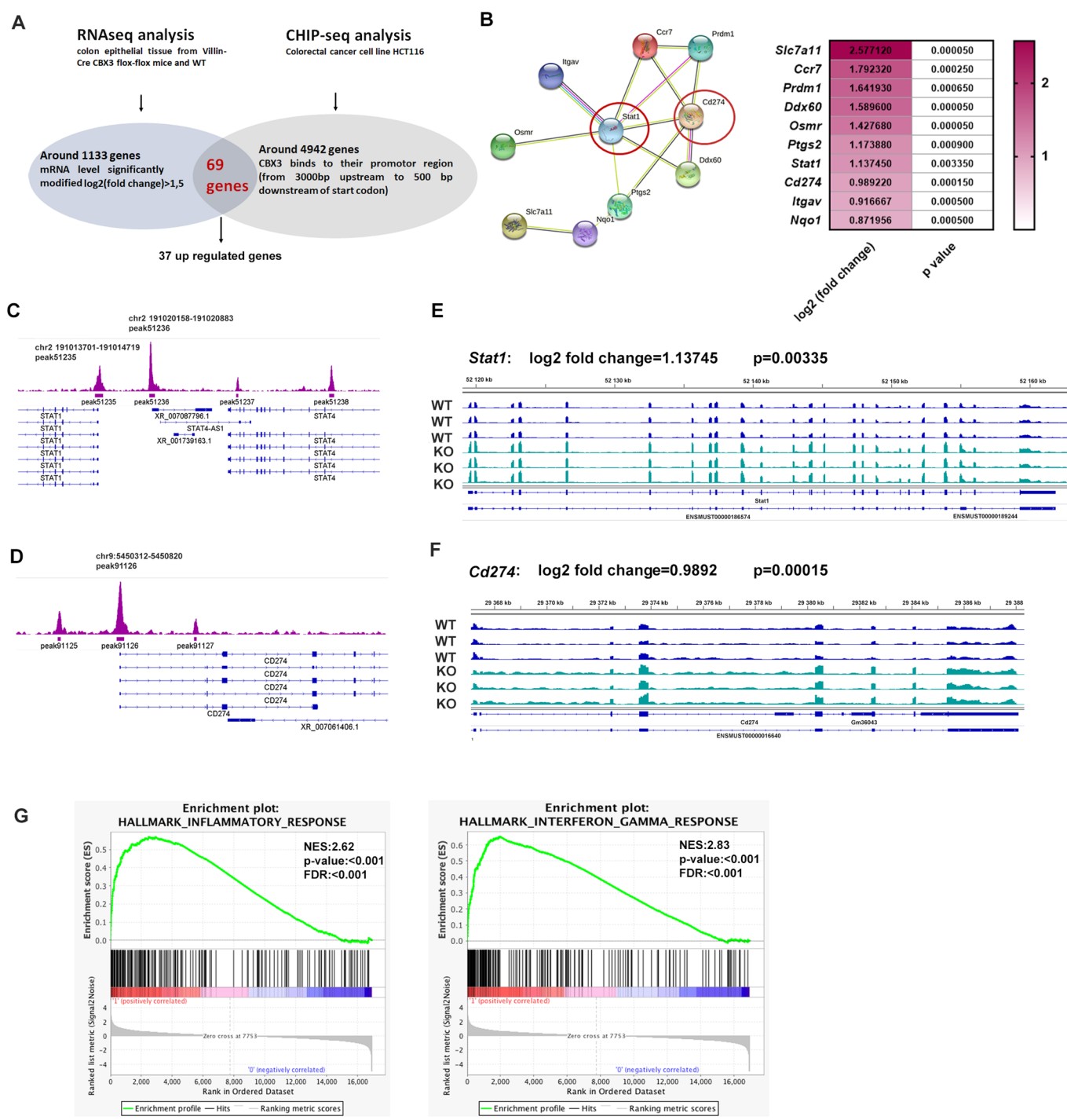

**Figure 3. Integrative analysis was applied to identify the direct gene target(s) of CBX3 in the colon epithelium.**

(**A**) Workflow showing CBX3 target genes identification with integrative analysis: RNA sequencing revealed 1133 genes with expression altered by more than 1.5-fold (increase or decrease) in *Cbx3* KO colon epithelium compared to WT. ChIP-seq identified CBX3 binding to the proximal promoter region of 4942 genes (from 3000 bp upstream to 500 bp downstream of the Transcription Starting Site). Crossing the two datasets led to the identification of 68 genes (37 upregulated and 31 down-regulated genes) as potential CBX3 target genes. (**B**) String analysis revealed a cluster of CBX3 direct targets involved in immune responses, including *Stat1* and *Cd274*. (**C, D**) ChIP-seq analysis identified two CBX3 binding sites around *STAT1* promoter region and one CBX3 binding site around *CD274* promoter region. (**E, F**) Integrative Genomics Viewer (IGV) was applied to visualize increased transcription levels of *Stat1* and *Cd274* in 3 *Cbx3* KO mice compared to 3 WT mice. (**G**) GSEA analysis based on transcriptome data revealed a significant enrichment of inflammatory and IFNγ responses in *Cbx3* KO colon epithelium. Signal2Noise uses the difference of means scaled by SD with FDR (false discovery rate) adjustment for multiple testing, $n = 3$.

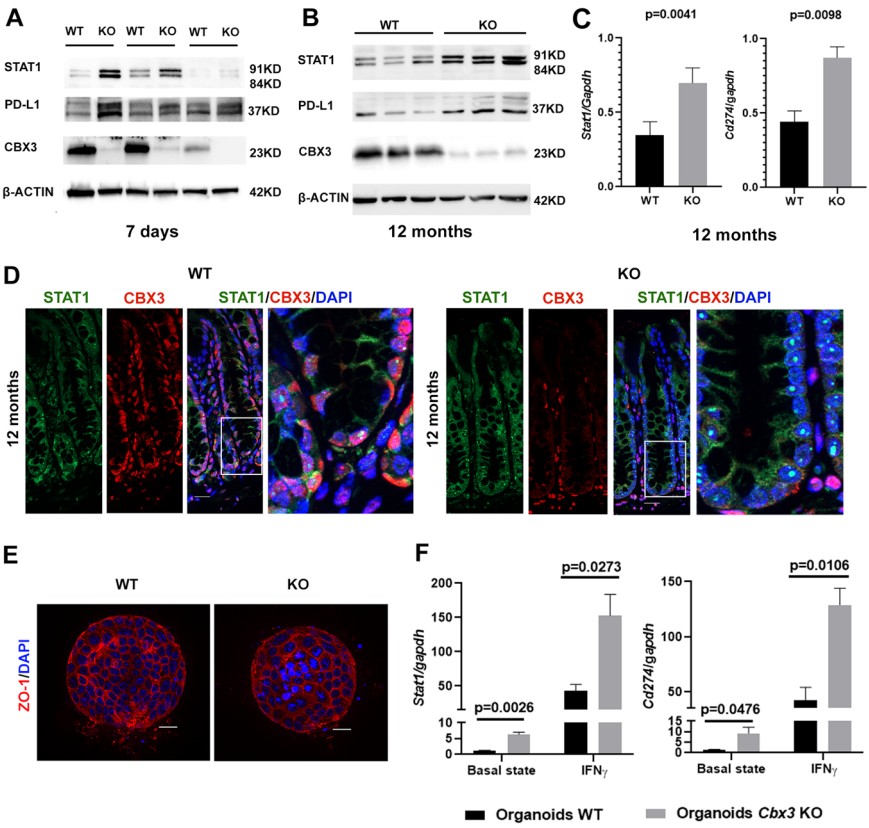

**Figure 4. CBX3 targets STAT1 and PD-L1 genes to repress in vivo and ex vivo their expression.**

(A, B) Western blot revealed increased STAT1 and PD-L1 protein levels in colon crypts 7 days (A) or 12 months (B) post-Tamoxifen gavages ($n = 3$ mice in each group). (C) RT-qPCR showing increased mRNA levels of Stat1 and Cd274 in colon epithelium 12 months post-tamoxifen induction ($n = 3$ mice in each group, two-sided t test). The error bar represented SEM. (D) Representative immunofluorescence of colon sections stained with anti-HP1γ (CBX3) antibody (red), anti-STAT1 (green), and DAPI (blue) in WT and Cbx3KO mice 12 months post-tamoxifen induction. The rightest figure represents the insert from its neighbor figure. Scale bar: 20 μm. (E) Organoids derived from Cbx3 KO and WT colon crypts were stained with anti-ZO-1 (red) to mark the cell-cell tight junction and DAPI (blue) to mark the nucleus. Scale bar: 20 μm. (F) RT-qPCR revealed that organoids derived from the Cbx3 KO colon expressed higher levels of Stat1 and Cd274 under basal culture state and under additional 40U IFNγ stimulation (3 mice in each group, two-sided t test). The error bar represented SEM. Source data are available online for this figure.

bidirectional *STAT1* promoter activity (Fig. 7A, red arrow). Q-PCR detection indicated that *CBX3* KO significantly increased LOC105373805 transcript level, which could be further enhanced 2 h after IFNγ stimulation (Fig. 7F). These results suggested that CBX3 depletion suppresses basal repression on *STAT1* promoter, which liberates it for further activation upon IFNγ stimulation.

Overall, these results are consistent with a model in which CBX3 binds at promoter regions to repress *STAT1* and *CD274* transcription. Upon IFNγ stimulation, CBX3 decreased its binding to the promoter which favorite gene expression.

### *CBX3* expression correlates negatively to *STAT1* or *CD274* expression in colorectal cancer

We then examined clinical data from colorectal cancer (CRC) patients to investigate the potential antagonist effect of CBX3 and IFNγ on the expression of *STAT1/PD-L1* in CRC. Clinical co-expression analysis based on colorectal adenocarcinoma was performed with the source data from TCGA (Cerami et al, 2012; Gao et al, 2013). The correlation between different gene expressions, such as *CBX3* VS. *STAT1*, *CBX3* VS. *CD274*, as well as *IFNγ* VS. *STAT1* or *IFNγ* VS. *CD274*, was based

on mRNA expression z-scores relative to diploid samples (RNA-Seq V2 RSEM, 592 samples). The patients were classified into two categories defined by the median of *IFNγ* or *CBX3* mRNA level. Interestingly, high *IFNγ* expression correlated with high *STAT1* or *CD274* expression in colorectal cancer, but high *CBX3* expression was significantly associated with low expression levels of *STAT1* or *CD274* (Fig. 8A).

Furthermore, in contrast to IFNγ which correlated positively with most members of JAK/STAT signaling, *CBX3* mRNA expression correlated negatively with the majority of JAK/STAT signaling genes' mRNA level, such as *JAK1*, *JAK2*, *STAT2*, and *STAT3*, etc. (Fig. 8B,C). These findings suggest that the antagonist effect between CBX3 and IFNγ may not be limited to STAT1/PD-L1, but rather to the entire JAK/STAT signaling pathway in colorectal cancer.

### Colorectal cancer patients with low *CBX3* expression exhibit better overall survival in considering both *CBX3* and *CD274* expressions

We then further investigated RNA-sequencing data from the TCGA colorectal cohort to identify the prognosis relation among *CBX3*, *STAT1*, and *CD274* expressions. Optimal cut points of

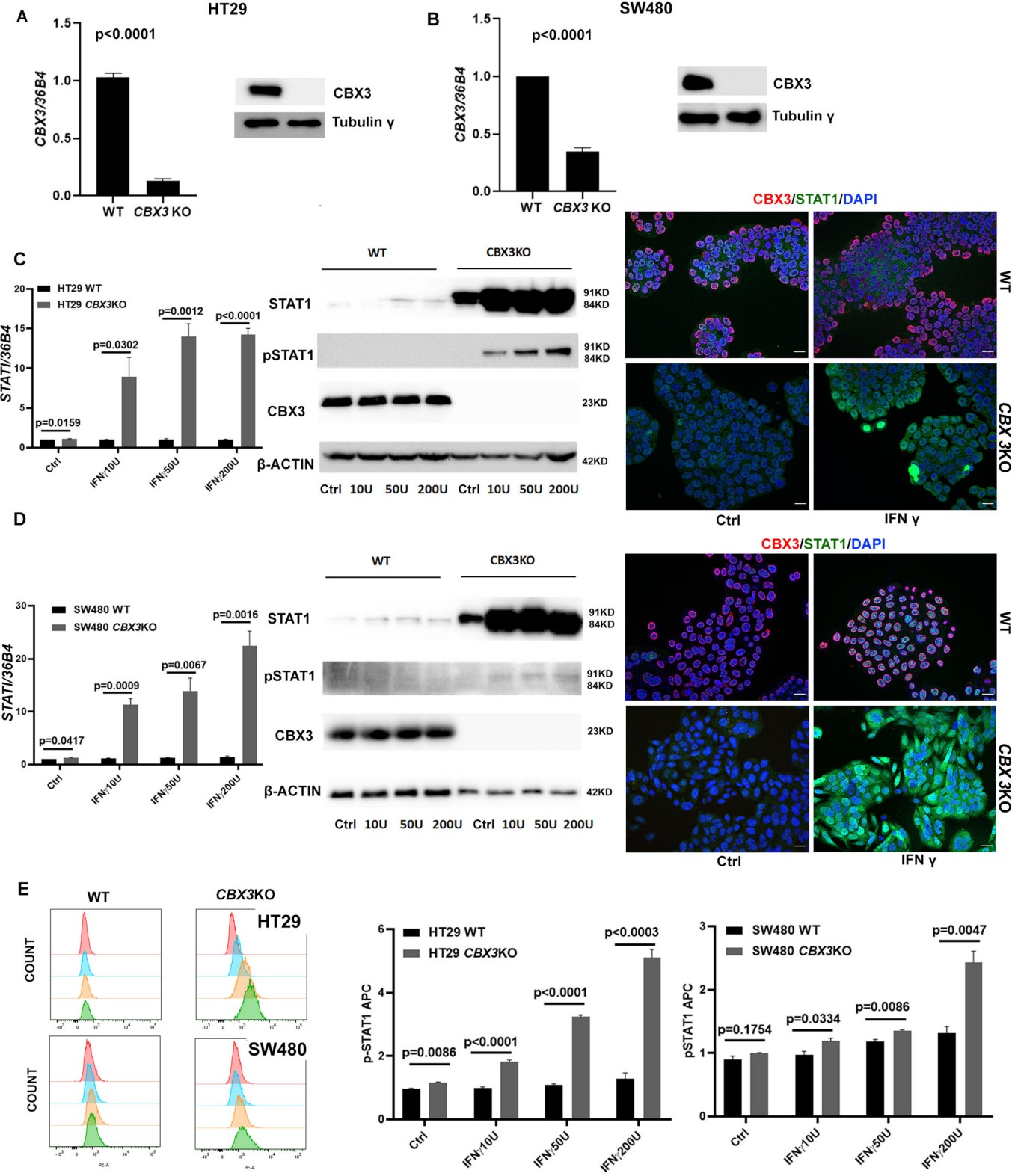

**Figure 5.   CBX3 deficiency significantly increases STAT1 in vitro expression upon IFNγ stimulation.**

(A, B) RT-qPCR and Western blot were performed to check CBX3 mRNA and protein levels after CRISPR/Cas9-mediated *CBX3* deletion in two CRC cell lines, HT29 and SW480. $n = 3$ independent experiments, two-sided t test for RT-qPCR results. The error bar represented SEM. (C, D) RT-qPCR (left), Western-blot (middle), and immunofluorescence (right) analysis revealed that IFNγ stimulation dramatically increased STAT1 expression in *CBX3* KO cells compared to WT controls. $n = 3$ independent experiments, two-sided t test for RT-qPCR results. The error bar represented SEM. Immunofluorescence was performed with anti-HP1γ (CBX3) antibody (red), anti-STAT1 (green), and DAPI (blue) for WT and KO cells without or with IFNγ stimulation (200U). Scale bar: 20 μM. (E) Cytometry analysis by using an anti-STAT1 Phospho (Tyr701) PE antibody was performed to check p-STAT1 level in *CBX3* KO cells. The IFNγ stimulation significantly increased p-STAT1 level in two different types *CBX3*KO cells (3 independent experiments, two-sided t test). The relative fluorescence intensity is presented as mean with SEM. Source data are available online for this figure.

expression with overall survival as outcome were determined for each investigated marker (Fig. 8D). Kaplan–Meier and log-rank analysis stratified on *CBX3*, *STAT1*, and *CD274* expression.

First, combined stratification based on *CBX3* and *STAT1* expression revealed a worse prognosis for patients with *CBX3*-high *STAT1*-low level expression as compared to all other CRC samples (Fig. 8E, left).

The situation for *CD274* is more complex. Combined stratification with *CBX3* and *CD274* expression revealed a significant relation of overall survival among four groups of samples: *CBX3*-high/*CD274*-low, *CBX3*-low/*CD274*-low, *CBX3*-high/*CD274*-high, and *CBX3*-low/*CD274*-high. First, the majority of the CRC patients presented a low CD274 expression (86.3% of total patients), *CBX3*-high/*CD274*-low group exhibited a longer overall survival compared to *CBX3*-low/*CD274*-low group (Fig. 8E, right). However, the *CD274*-high group (only 13.7% of total patients) showed a lower overall survival notwithstanding the high or low expression level of *CBX3*.

Taken together, these results suggest that expression levels of *CBX3*, *STAT1*, and *CD274* have opposite prognostic significance for CRC patients. High expression of *CBX3* is generally associated with a worse prognosis, particularly when it is associated with a low expression of *STAT1*. Moreover, low expression of *CBX3* is indicative of a better prognosis for the majority of CRC patients who exhibit a low expression of *CD274*.

### *CBX3* KO sensitizes CRC cells to chemotherapy through regained IFNγ sensitivity

Chemotherapy is one of the standard treatment strategies for CRC at various stages. Since *CBX3* KO can render IFNγ-insensitive CRC cells becoming dramatically sensitive to IFNγ stimulation, we further investigated whether this restored sensitivity to IFNγ could make *CBX3* KO CRC cells more sensitive to chemotherapy.

Firstly, the effect of CBX3 deletion and IFNγ stimulation on cell survival of HT29 and SW480 cells was examined. *CBX3* KO led to higher cell proliferation in HT29 cells. Ki67 and Annexin labeling indicate *CBX3* deletion led to higher cell proliferation in HT29 cells (Fig. EV4A) but makes HT29 more sensitive to IFNγ induced apoptosis (Fig. EV4B). However, these effects are not observed in SW480 cells (Figs. 9A,B and EV4).

Despite the different effects on the proliferation of different CRC cells, *CBX3* KO significantly increased or tended to increase the chemosensitivity of both HT29 and SW480 cells to Irinotecan and Fluorouracil (5-FU), two of the first-line chemotherapy drugs in CRC treatment (Fig. 9C,D). Furthermore, the sensitivity of *CBX3* KO cells to Irinotecan or to 5-FU remarkably increased after IFNγ stimulation (applied 24 h before and 48 h during the drug treatment). More specifically, for SW480 *CBX3* KO cells, this effect

was particularly pronounced in response to 5-FU treatment while for HT29 *CBX3* KO cells, this effect was evident for both treatments.

Further analysis to compare the cell viability with or without additional IFNγ stimulation revealed that IFNγ stimulation increased only CBX3 KO cells' chemosensitivity and had little effect on WT CRC cells (Fig. 9E,F).

All together, these results show that IFNγ stimulation can remarkably sensitize CBX3 KO CRC cells to chemotherapy, suggesting that the increased sensitivity to IFNγ in *CBX3* KO cells can deeply increase their chemosensitivity.

### *Cbx3*KO increased the chemosensitivity of mice CRC tumors under IFNγ stimulation

We finally confirmed that *Cbx3* KO can increase CRC chemosensitivity under IFNγ stimulation with MC38 syngeneic mouse tumor model.

We first generated CRISPR/Cas9 *Cbx3* KO MC38 cell. *Cbx3* KO increased the sensitivity of MC38 cells to IFNγ, indicated by the increase of STAT1 expression under IFNγ stimulation (Fig. 9G). At the same time, *Cbx3* KO increases also the chemosensitivity of MC38 cells to 5-FU combined to IFNγ (Fig. 9H).

We then grafted MC38 cells (WT or *Cbx3* KO) to C57BL/6 mice. Seven days after implant, the mice with WT or *Cbx3* KO tumors were then treated by 5-FU or 5-FU combined to IFNγ until 22nd day (Fig. 9I). Tumors' weight at the end of treatment are presented in Fig. 9J. Tumors' volumes were also quantified 3 times each week to compare the growth rate between WT and *Cbx3* KO tumors under different treatments (Fig. 9K).

The size of WT and *Cbx3* KO tumors without treatment exhibited no significant difference, even the KO tumors seem showing a tendency to grow bigger than WT tumor around 21st and 22nd days (Fig. 9J,K). Despite of this tendency, the deletion of *Cbx3* significantly increased MC38 tumors' chemosensitivity to 5-FU combined IFNγ compared to WT MC38 tumors (Fig. 9J,K).

## Discussion

In this study, we have identified a novel role of CBX3 in antagonizing the IFNγ signaling cascade in the colon epithelium. We found that CBX3 represses the transcription of STAT1 and CD274 to antagonize IFNγ stimulation. Likewise, upon IFNγ stimulation, we observed a decreased CBX3 binding to the promoters of STAT1 and CD274, concomitant with their increased gene expression. Importantly, depletion of CBX3 led to a dramatic increase of STAT1 and PD-L1 expression under IFNγ stimulation, strongly suggesting that CBX3 is an important checkpoint to

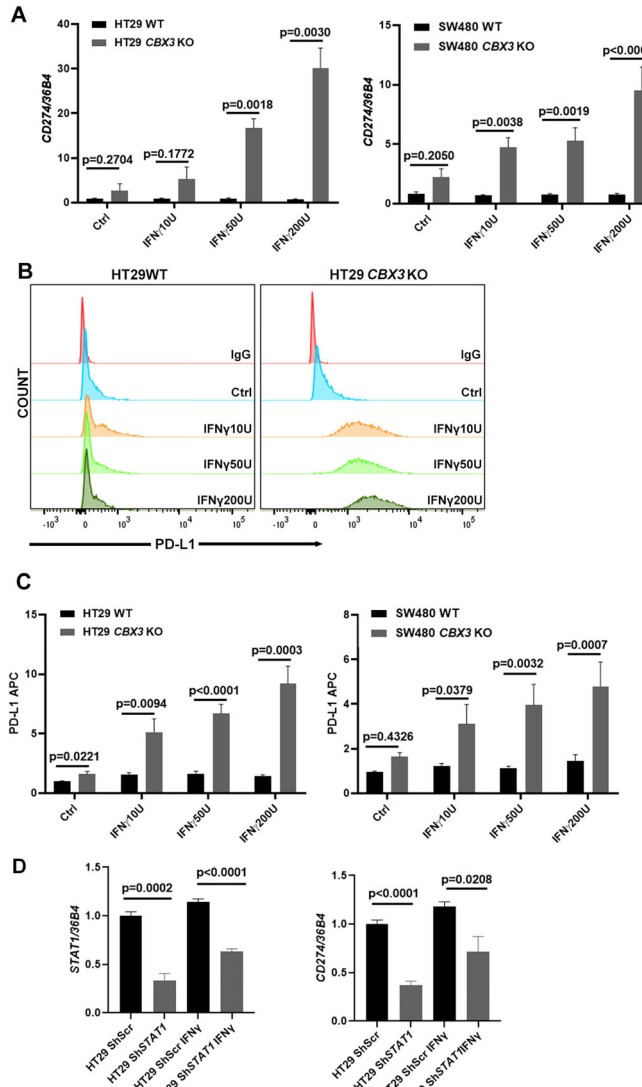

**Figure 6. CBX3 deficiency significantly increases PD-L1 in vitro expression upon IFNγ stimulation.**

(A) *CD274* mRNA expression strongly increased in HT29 and SW480 *CBX3* KO cells compared to WT upon IFNγ stimulation. Three independent experiments, two-sided Student's t-test. The error bar represented SEM. (B, C) Flow Cytometry detected cell surface PD-L1 expression. HT29 and SW480 KO cells significantly increased cell surface PD-L1 expression compared to WT upon different concentrations of IFNγ stimulation. (B) Representative staining of HT29 WT and *CBX3* KO cells is shown. (C) Different histograms display the mean values of PD-L1 APC fluorescence intensity of WT and KO cells under different experimental conditions. Three independent experiments, two-sided Student's t test. The error bar represented SEM. (D) Knock down *STAT1* by shRNA in HT29 cells significantly deceased *CD274* expression either in basal condition or under IFNγ stimulation. Three independent experiments, two-sided Student's t test. The error bar represented SEM. Source data are available online for this figure.

control IFNγ stimulated immune gene activation. Interestingly, deletion CBX3 in SW480 or HT29 cells is enough to induce dramatic STAT1 and CD274 expression under IFNγ stimulation, even though there exists a compensatory increase of CBX5 (HP1α) mRNA and CBX1 (HP1β) mRNA is identified in SW480 cells.

CBX3 antagonist effect to IFNγ could not be compensated by other members of HP1 family.

Overall, our model predicts that CBX3 is required for fine-tuning the expression of IFNγ-stimulated immune genes, which makes CBX3 implicated in modulating colon inflammatory response as well as colon cancer chemo-resistance.

As we presented in the introduction, aberrant IFNγ signaling is associated with ulcerative colitis, a chronic inflammatory bowel disease (IBD). Our study here reveals that CBX3 keeps the IFNγ signaling in check by repressing *STAT1/CD274* transcription in the colon epithelium, suggesting the potential role of CBX3 in maintaining colon immune homeostasis. Consistently, knockout led to chronic inflammation in the mouse colon, even without additional external stimulus. Long-term deletion of CBX3 further induced lymphoid nodule hyperplasia which has been reported in association with IBD (Colarian et al, 1990; Kenney et al, 1982). Our previous study also reported that CBX3 expression was reduced in UC patients and CBX3 regulates RNA splicing precision, this homeostatic function in RNA metabolism being altered in IBD (Mata-Garrido et al, 2022). Taken together, these accumulated data suggested that Villin-creERT2:Cbx3−/− mice could be a very attractive model for further exploring the potential link between dysregulated inflammatory/immune response and UC development.

Moreover, our clinical analysis based on CRC patients revealed that CBX3 correlated negatively with almost all the members of JAK/STAT signaling, including JAK1, JAK2, STAT1, STAT2, and STAT3. Upadacitinib, a novel JAK inhibitor, significantly increased the proportion of active ulcerative colitis patients who achieved remission when compared with placebo (26% versus 5% and 36% versus 4% in the two cohorts) (Kobayashi and Hibi, 2023). The antagonist role of CBX3 to JAK/STAT signaling suggest that CBX3 could also be an interesting therapy target to improve IBD treatment in the future.

Our clinical analysis based on TCGA CRC patients' mRNA levels confirmed the antagonist effect between *CBX3* and *IFNγ* on *STAT1* and *CD274* expression in CRC. Indeed, high *CBX3* expression was associated with the low expression of *STAT1/CD274*, but high expression of IFNγ was correlated with high expression of *STAT1/CD274*. The negative correlation between the expression of *CBX3* and *STAT1* or *CD274* is not as strong, which could be due to the increased expression of *CBX3* in CRC tissues compared to normal tissues (Li et al, 2020; Liu et al, 2015). Given the important role of IFNγ/STAT1/PD-L1 axis in oncogenesis and cancer treatment, targeting the IFNγ signaling pathway as therapy is a rational and novel management in colorectal cancer (Du et al, 2022). Our findings indicate that CBX3 could antagonize IFNγ, CBX3 thus becomes a potential target to improve colorectal cancer treatment. In fact, *CBX3* KO makes IFNγ-insensitive CRC cells become radically sensitive to IFNγ, this modification considerably improves the efficacy of the two oldest and widely used CRC chemotherapy drugs: Irinotecan and 5-FU.

The effect of CBX3 knockout on chemosensitivity is cell-line dependent. Specifically, under IFNγ stimulation, SW480 KO cells become particularly sensitive to 5-FU while HT29 KO cells are relatively more sensitive to Irinotecan treatment. This difference could be due to many factors, as their different origin, mutation statuses, and their different genetic and epigenetic features, among others. (Rochette et al, 2005; Yeh et al, 2009; Ren et al, 2017;

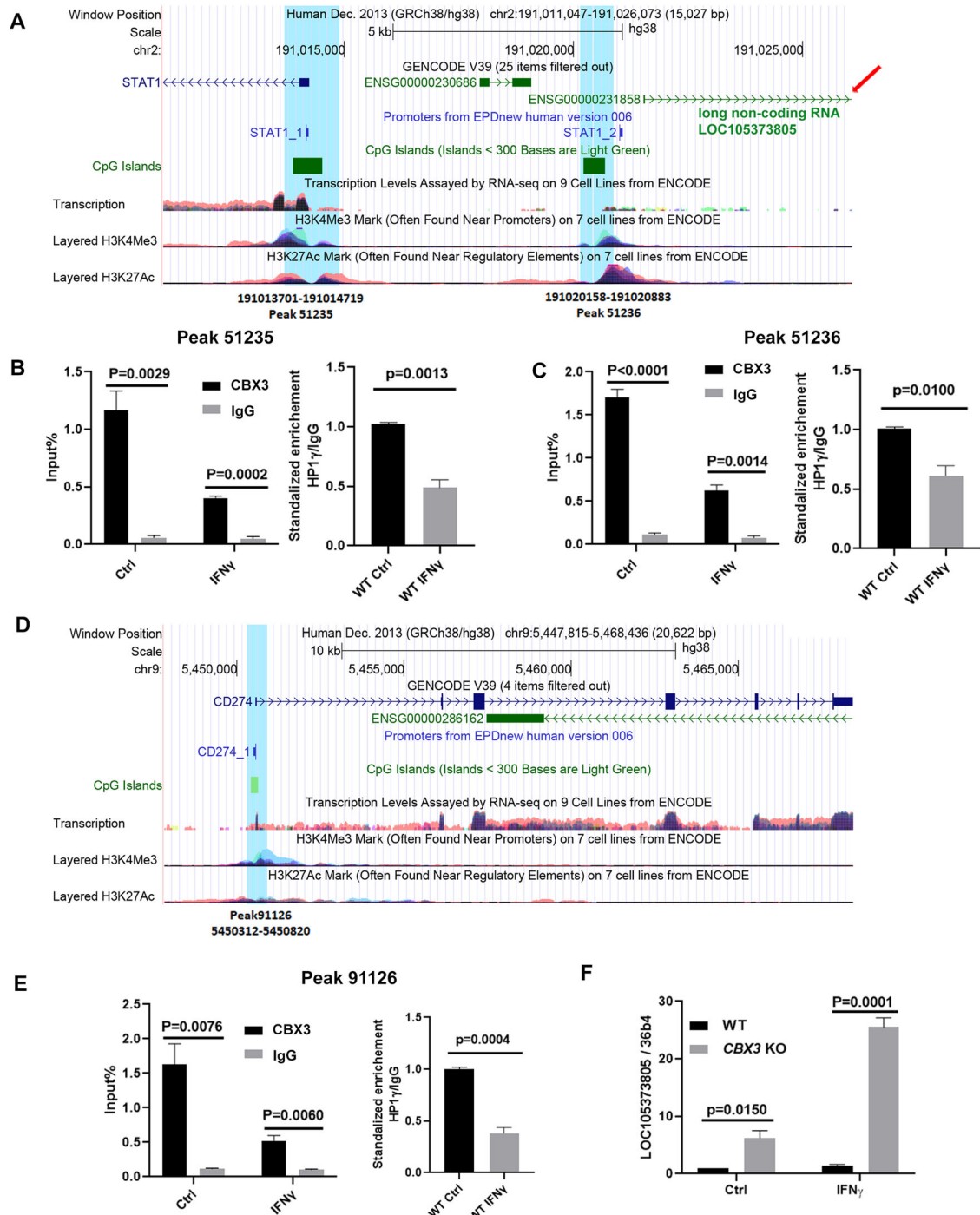

Ahmed et al, 2013; Tate et al, 2019; Mitsopoulos et al, 2021). All these factors ultimately lead to their different level of STAT1 phosphorylation after IFNγ stimulation (Fig. 5C,D, middle panels, as shown by Western blot). Higher phosph-STAT1 signifies a greater STAT1 transcriptional activity, which could further amplify IFNγ signaling to induce higher expression of death-promoting genes (Kim and Lee, 2007) and ultimately increase the sensitivity of HT29 CBX3 KO cells to IFNγ (Fig. 9A) as well as to chemotherapy under IFNγ stimulation (Fig. 9C,D).

Furthermore, Kaplan–Meier and log-rank analysis showed that low CBX3 but higher STAT1 expression is associated with a better overall survival of CRC patients. Consistent to our findings, several previous clinical studies have also reported that high CBX3 expression is associated with reduced disease-free survival rates and CRC progression (Li et al, 2020; Liu et al, 2015). As the chemotherapy based on 5-FU, Irinotecan, Oxaliplatin etc. remains for almost a decade as the most effective current anti-CRC therapy, these clinical findings suggest that low CBX3 expression may favor

**Figure 7. IFNγ stimulation reduces the binding of CBX3 to the promoters of *STAT1* and *CD274*.**

(A) UCSC Genome Browser centered on *STAT1* gene: The blue labeling corresponds to the ChIP-seq Peaks 51235 and 51236. These CBX3 binding sites are located at the promoter region of *STAT1* gene enriched by H3K4Me3, H3K27Ac, CpG islands. The red arrow designates the non-coding RNA LOC105373805, upstream of the *STAT1* promoter and in opposite orientation. (B, C) ChIP experiments were carried out in HT29 cells with and without IFNγ stimulation. The antibody against CBX3 (HP1γ) or IgG were applied for chromatin immunoprecipitation. CBX3 chromatin association was detected by Q-PCR with specific primers to Peak 51235 and 51236 regions of *STAT1* gene. The left panel represents input% of CBX3 and IgG antibodies and the right panel represent normalized enrichment (Input% of CBX3 antibody compared to Input% of IgG). Three independent experiments, two-sided Student's t-test. The error bar represented SEM. (D) UCSC Genome Browser centered on *CD274* gene: ChIP-seq Peak 91126 for *CD274* gene is labeled in blue, matching with its promoter regions, indicated by an enrichment of H3K4Me3, H3K27Ac, and CpG islands. (E) ChIP experiments were carried out with the antibody against CBX3 or IgG. CBX3 chromatin association was detected by Q-PCR with specific primers to Peak 91126 region of *CD274* gene. The left panel represents the input% of CBX3 and IgG antibodies, the right panel represents the normalized enrichment (Input% of CBX3 antibody compared to Input% of IgG). Three independent experiments, two-sided Student's t-test. The error bar represented SEM. (F) The transcript level of LOC105373805 was significantly increased in HT29 CBX3 KO cells, 2 h IFNγ (200U) stimulation could further enhance it. Three independent experiments, two-sided t test. The error bar represented SEM. Source data are available online for this figure.

the efficacy of chemotherapy in CRC patients. *Cbx3* KO increased CRC tumor chemosensitivity under IFNγ stimulation is also confirmed with MC38 syngeneic mouse tumor model. Therefore, these clinical insights, in conjunction with our in vitro and in vivo studies, underscore the relevance of CBX3 as a potential therapeutic target for future clinical CRC chemotherapy treatments.

Another key point is that CBX3 deletion also significantly increased PD-Ll expression under IFNγ stimulation. PD-1 and its ligand PD-L1 are two important targets for immune checkpoint blockade (ICB) therapy (Dong et al, 2002). However, until now, only a small subset of dMMR-MSI-H (mismatch repair-deficient and microsatellite instability-high) CRC patients (less than 15%) benefit from ICB therapy (Puccini et al, 2020; Du et al, 2022). Generally, high PD-L1 expression, as well as upregulated IFNγ signaling, is believed as a favorable element to anti-PD-1/PD-L1 therapy (Yi et al, 2018; Du et al, 2022; Alspach et al, 2019). CBX3 deletion makes CRC cells regain IFNγ sensitivity and dramatically increased PD-L1 expression in two MSS (Microsatellite Stable) and MMR (Mismatch repair) proficient CRC cell lines (SW480 and HT29). These attractive results suggest that CBX3 could be a potential therapeutic target to improve ICB therapy for the majority of CRC patients. This topic certainly deserves further investigation in the future.

In conclusion, our work depicts a fascinating interplay between CBX3 and IFNγ signaling in regulating STAT1 and PD-L1 transcription. It opens up attractive perspectives for exploring the role of CBX3 in bowel inflammation response as well as its clinical potential in CRC chemo- or immuno-therapies.

# Methods

## *Cbx3* KO mouse model

The Villin-creERT2:*Cbx3*[-/-] mouse model and Tamoxifen administration were produced as previously described (Mata-Garrido et al, 2022). Briefly, Tamoxifen (10 mg/mouse, calculated based on 0.5 mg/g of body weight, with each mouse weighing approximately 20 g) diluted in 20% clinOleic acid was administrated by oral gavage, at 3 doses every 5 days in Cbx3Flox/Flox;Tg(Villin-CreERT2) mice, noted as *Cbx3* Villin-Cre or *Cbx3* KO mice in the later text. Control Cbx3 Villin-Cre mice received 20% clinOleic acid alone by oral gavage. Additional controls using Cbx3Flox/Flox mice that do not express the Cre recombinase were identically treated with Tamoxifen (Fig. EV5). Animal studies were approved by the ethical committee of Paris Descartes University

(authorization number 17-022). The mice were housed in a maximum of 5 ventilated cages in accordance with their social needs, with water and food provided ad libitum. To ensure the best possible housing conditions for the animals, cardboard houses and tunnels, as well as wooden sticks for gnawing, were added to the cage to reduce stress and anxiety for the mice. The mice were monitored daily.

## Colon crypts isolation and organoids culture

Colon crypt was isolated as previously described at 7 days and 12 months post-tamoxifen induction (Mata-Garrido et al, 2022). *Cbx3* KO and WT mice colon organoids were cultured in Matrigel and after 3 days of culture, the mouse 40 U/ml IFNγ (Miltenyi) was added to the culture medium for an additional 24 h before RT-qPCR analysis.

## Cell culture and *CBX3* KO cell lines construction

Human SW480 and HT29 cells were originated from ATCC (CCL-228 and HTB-38). MC38 cell was kindly provided by Dr. Eliane Piaggio. The cells were cultured in Dulbecco's modified Eagle medium (DMEM) supplemented with GlutaMAX (GIBCO, Life Technology) and 10% fetal bovine serum (FBS, Hyclone) with 5% $CO_2$. No mycoplasma contamination was detected. To generate CRISPR/Cas9-mediated *CBX3* KO cell line, HT29, SW480 and MC38 cells were transfected in 6-well plates at around 80% confluency in the presence of Lipofectamine 2000 (Life Technologies). 1 µg Cas9+sgRNA plasmid (Mata-Garrido et al, 2022) was used for each well. Single GFP+ cells were sorted by cytometry to 96-well plates 48 h after transfection. *CBX3* deletion in CRISPR/Cas9 *CBX3* KO cell lines derived from different single-cell clones were confirmed by Western blot and RT-qPCR before experiments.

For *STAT1* knockdown, HT29 cells were seeded in a 48-well plate, and then transfected for 24 h with 50 µl lentiviral particles (SIGMA, NM_007315/TRCN0000004265) when cells arrived at 70% confluence. The cells stay in DMEM, 10% FBS without puromycin for 2 days and then transplanted in a six-well plate. After the cell adhesion, 2 µg/mL puromycin (Thermo Fisher) was added in the culture milium for 1 week. The stable shRNA *STAT1* cell line was used for further analysis.

## RT-qPCR

RT-qPCR for cells was performed as previously described (Zhang et al, 2020). For organoids RT-qPCR analysis, the organoid

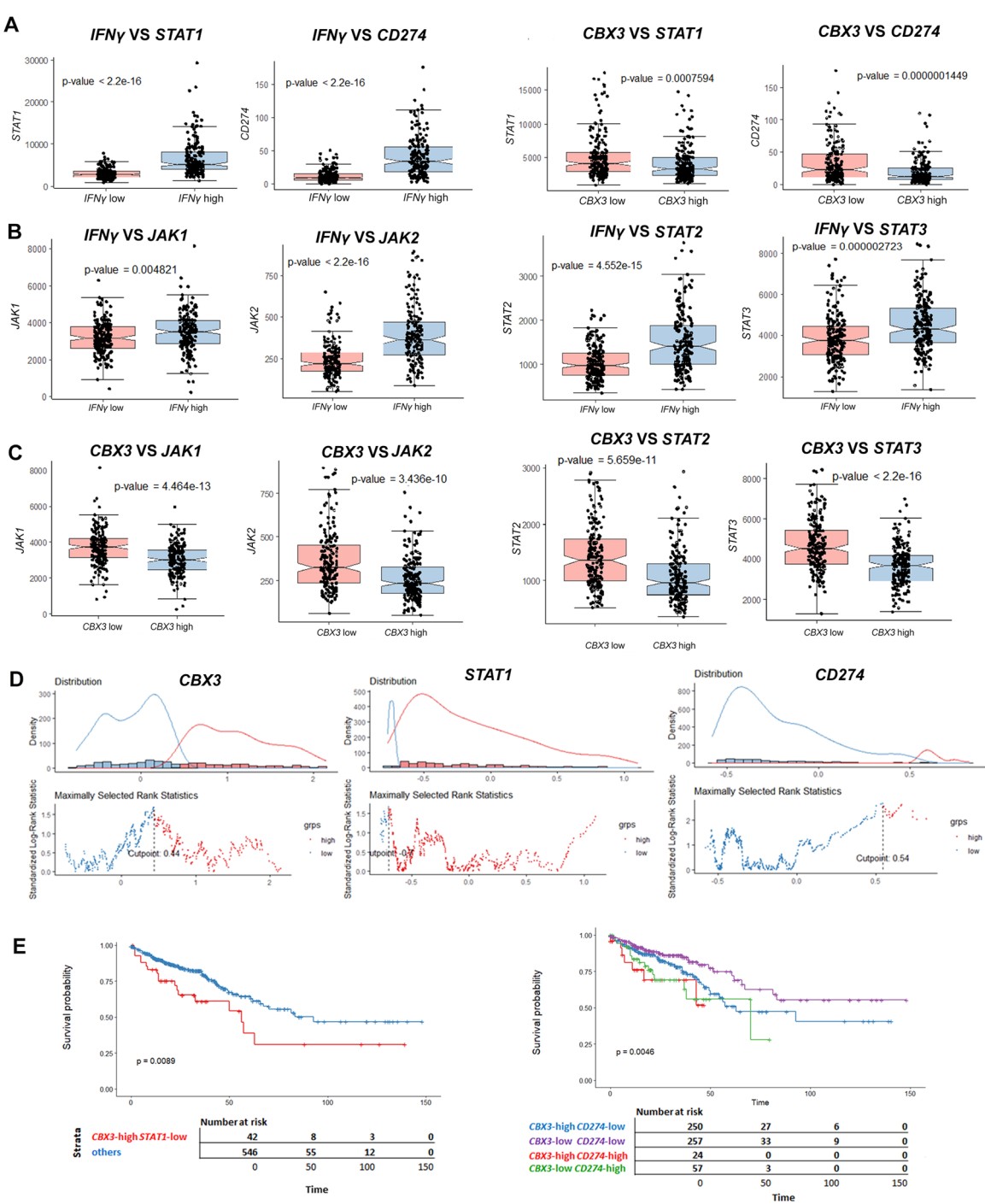

culture dish was incubated 20 min on ice to dissolve Matrigel before collecting the sample. Primers used for samples are listed in Table EV1. *36B4* or *Gapdh* were used as a reference for target gene quantification with the "delta delta Ct" (ΔΔCt) method.

## Western blot

Western blot were performed with the CRC cells samples as previously described (Zhang et al, 2020). Mouse colon crypts were isolated and lysed in RIPA buffer (50 mM HEPES 0.1% SDS, 1% Triton X-100, 1 mM EDTA, 0.5% sodium deoxycholate, 150 mM NaCl) with proteinase inhibitor (cOmplete™, EDTA-free Protease Inhibitor Cocktail, Roche). Western blots were performed with following antibodies: Anti-STAT1 (14994, Cell Signaling, 1:1000), anti-phosphorylated STAT1 (9167, Cell Signaling, 1:200 or 1:1000), anti-HP1γ (IG-2MOD-1G6-AS, Euromedex, 1:2000), anti-PD-L1 (4059, Prosci, 1:1000), anti-β-actin (A3854, Sigma, 1:25,000 or 1:50,000), anti-Tubulin (4D11, Thermo Scientific, 1:1000).

◀ **Figure 8. Clinical analysis revealed the negative correlation and opposite prognostic significance of *CBX3* and *STAT1/CD274* expression for CRC patients.**

(A) Clinical co-expression analysis in CRC patients was based on mRNA expression z-scores relative to diploid samples (RNA Seq V2 RSEM, 592 samples). The high and low categories are cut on the median of the expression level of *CBX3* and *IFNγ*, each group is composed of 296 tumor samples. The analysis showed that high *IFNγ* expression correlated to high expression of *STAT1* or *CD274*, while high *CBX3* expression associated with low expression of *STAT1* or *CD274*. Two-sided t test was applied for statistical analysis. Box of boxplot represents interquartile (IQR) Q1-Q3 (50 percent of samples) and error bar represent 1.5 times IQR allowing exclusion of outliers. (B, C) Clinical co-expression analysis based on Colorectal Adenocarcinoma revealed that IFNγ correlated positively to *JAK1*, *JAK2*, and *STAT2*, while *CBX3* correlated negatively to *JAK1*, *JAK2*, and *STAT2*. Two-sided t test was applied for statistical analysis. Box of boxplot represents interquartile (IQR) Q1-Q3 (50 percent of samples) and error bar represent 1.5 times IQR allowing exclusion of outliers ($n = 296$). (D) The Optimal cut points of expression with overall survival as outcome were determined by R software for *CBX3*, *STAT1*, and *CD274* expression in CRC tumors (RNA-sequencing transcriptome experiments from the TCGA colorectal cohort). (E) Left: Overall survival analysis was stratified based on combined *CBX3* and *STAT1* expression thresholds. Colorectal cancer patients with *CBX3*-high *STAT1*-low samples exhibited worse overall survival. Right: Overall survival analysis was stratified based on combined *CBX3* and *CD274* expression thresholds. Low *CBX3* expression indicates a better overall survival for the majority CRC patients who exhibit a low expression of *CD274*. For all the overall survival analysis, time is represented in months. The patients' number associated with each group is showed in the under table. Tables under Kaplan–Meier graph represent patients still at risk for the distinct times of follow (0, 50, 100, 150 months) according the molecular group stratification. The log-rank test of the Cox proportional Hazard model was applied for all the overall survival analysis. Source data are available online for this figure.

## Immunofluorescence and immunochemistry

Immunofluorescence (IF) were performed as previously described (Zhang et al, 2020; Mata-Garrido et al, 2022). The organoids were grown on coverslip and fixed with 4% paraformaldehyde at 4 °C overnight before experiments. IF were performed with the following antibodies: Anti-STAT1 (14994, Cell Signaling, 1:100), anti-ZO-1 (Invitrogen, 33-9100, 1:100), and Anti-HP1γ (IG-2MOD-1G6-AS, Euromedex, 1:100).

For immunochemistry, slides were processed on the automaton Leica Bond RX and unmasked at pH 6 (except pH 9 for CD68) before being incubated 30 min with an anti-CD8 antibody (Abcam, ab98941, 1:200) or anti-CD4 (Abcam, Ab183685, 1:500), anti-Ly6G (Biolegend 127605, 1:400), anti CD68 (Abcam, Ab12512, 1:400), anti-Ki67 (Abcam15580, 1:200), and anti-BrdU (Sigma B8434, 1:1000) then washed. BrdU (Sigma) was injected intraperitoneally at 100 μg/g animal body weight, 1 h prior to sacrifice. The revelation system ("Bond Polymere Refine" kit, DS9800, Leica) included a secondary antibody HRP (Cell Signaling, 98941, 1:1000 to 1:3000). Hematoxylin (blue) counterstaining allows the visualization of cell nuclei. CD4+ or CD8+ T cells, CD68+ macrophage, Ly6G+ neutrophil, Ki67+ and Brdu+ cells were counted with 4 μm thickness colon tissue section with around 10–20 sections from 4 mice of each distal or proximal group.

## ChIP-qPCR

ChIP experiments were performed as previously described (Batsché et al, 2006; Saint-André et al, 2011). Clarified samples were incubated overnight with HP1γ (CBX3) antibodies (Sigma-Aldrich, 05-690) or mouse nonimmune IgG as negative control. Immuno-precipitated DNA was quantified by qPCR with primer sets specific for the indicated CBX3 binding domain (Table EV2). Reference background levels were estimated by ChIP with nonimmune IgG. qPCR with primer sets aside from for indicated CBX3 binding domain were used as negative control. The relative quantification (Enrichment) was calculated with the ratio between ChIP with CBX3 and with nonimmune IgG.

## ChIP-seq and RNA-seq integrative analysis

The CBX3 ChIP-seq dataset on HCT116 (GSE28115) (Data ref: Smallwood et al, 2012) was mapped onto the human genome

(hg38) using the Binding and Expression Target Analysis (BETA) python algorithm. This allowed for the identification of proximal binding sites around −3000 and +500 base pairs from the Transcription Stating Sites (TSS) of human genes (Wang et al, 2013). Gene set enrichment analysis on RNA-seq from GSE192800 (Data ref: Mata-Garrido et al, 2022) was performed by using GSEA version 4.3.0 standalone application (Subramanian et al, 2005). During the integrative analysis, CBX3 targets identified by ChIP-seq and upregulated in the RNA-seq were used as input for network analysis in the STRING web tools version 11.5 (Szklarczyk et al, 2021).

## Flow cytometry analysis

For cell surface labeling, cells were detached with Trypsin-EDTA (Thermo Fisher, MA, USA), centrifuged and suspended in PBS containing 0.5% BSA, 2 mM EDTA and APC anti-human CD274 (Biolegend, 329708, 1:20) at 4 °C for 30 min. For apoptosis analysis, the APC Annexin V Apoptosis Detection Kit with 7-AAD (Biolegend, 640930, APC Annexin V, 1:20) was used with following the manufacturer's instructions. For Ki67 and p-STAT1 labeling, cells were harvested, then fixed with cold 80% ethanol, incubated at −20 °C for minimum 2 h, then labeled either with APC Ki67 (Milteny Biotec, 130100330, 1:50) and PI (50 μg/mL) or with PE anti-STAT1 Phospho (Tyr701) (Biolegend, 666403, 1:10), respectively. After labeling, cells were washed and analyzed by LSR Fortessa™ cell analyzer (Becton Dickinson, NJ, USA).

## Bioinformatics clinical survival analyses and co-expression analyses based on colorectal cancer RNA-seq transcriptome

TCGA Matrix of colorectal cancer RNA-sequencing V2 Z score was downloaded on Cbioportal website (https://www.cbioportal.org/) (Cerami et al, 2012; Gao et al, 2013; Liu et al, 2018) and aggregated with clinical data of the respective patients. This cohort of colorectal cancer comprised of 592 tumor samples (Dataset EV1). Overall survival time in months and status were integrated with the RNA-seq data to perform survival analyses in R software environment version 4.1.3 with survival R-package version 3.2-13 and Survminer R-package version 0.4.9. The optimal cut point of *CBX3*, *STAT1*, and *CD274* expressions were selected using

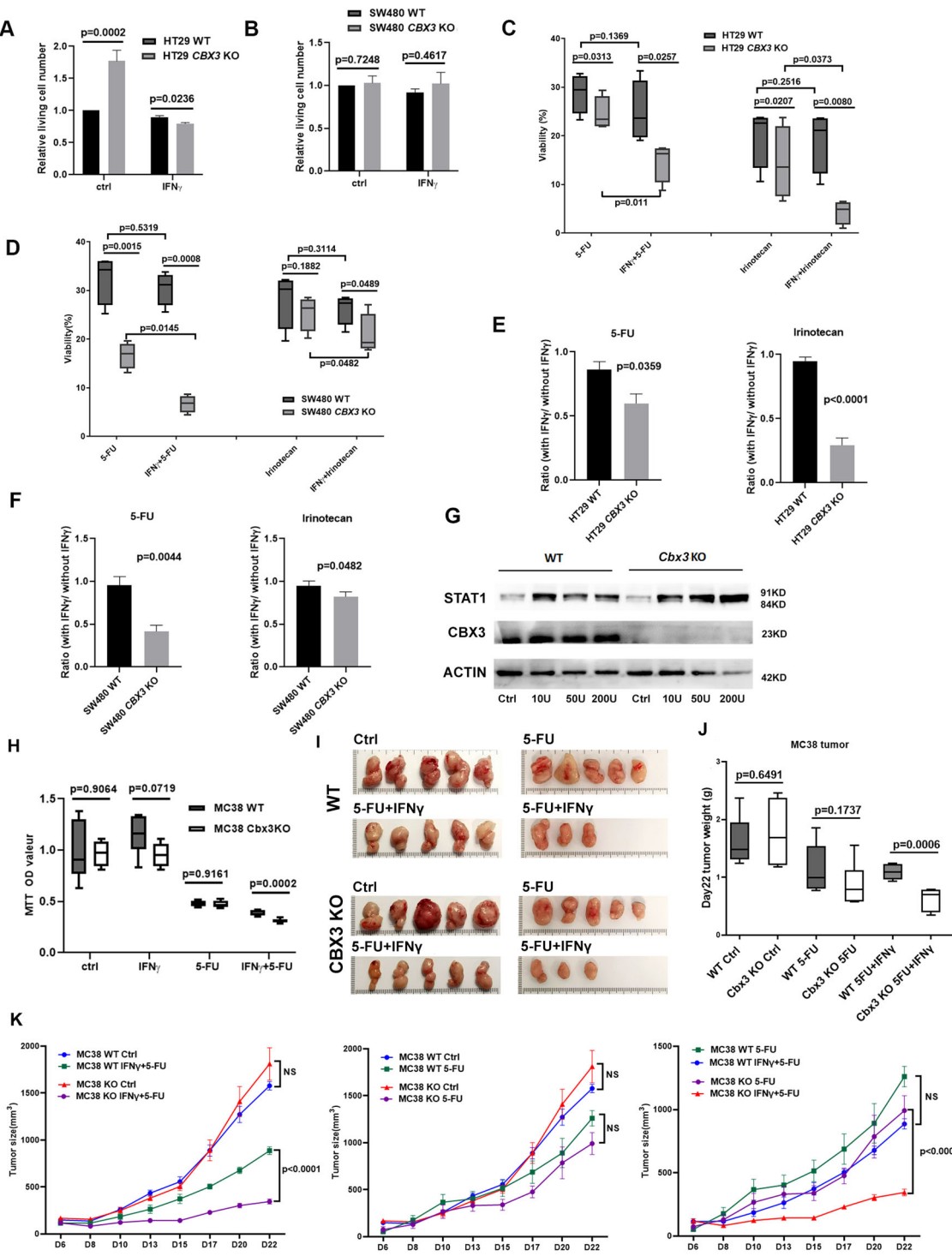

the Survminer package based on standardized log-rank statistics and Kaplan–Meier plot. The log-rank test was stratified based on their respective expression threshold.

The co-expression correction between different genes was based on mRNA expression z-scores relative to diploid samples (RNA-Seq V2 RSEM, 592 samples), which are classified in two categories defined on the median of $IFN\gamma$ or $CBX3$ mRNA expression level.

## Drug sensitivity assay

$20 \times 10^3$ HT29 cells or $40 \times 10^3$ SW480 cells per well were seeded in 12-well plate. After 72 h of culture, 200 U/ml human $IFN\gamma$ was added in culture medium. Twenty-four hours later, 100 μM Irinotecan or 100 μM 5-FU was added for an additional 48 h before analysis. The living cells with different treatments were counted by using a cell counter (Bio-Rad TC20TM) as Trypan blue

**Figure 9. *CBX3* KO sensitizes CRC cells to chemotherapy through regained IFNγ sensitivity.**

(A, B) CBX3 deletion and IFNγ stimulation produced different influence on cell proliferation of HT29 and SW480 cells. Living cell number counting revealed that *CBX3* KO increased cell proliferation in HT29 cells, but after 72 h IFNγ, less viable cell was counted in *CBX3* KO cells compared to WT control. These effects were not observed with SW480 cell The error bar represented SEM. (5 independent experiments, two-sided t test). (C, D) *CBX3* KO significantly increased the chemosensitivity of HT29 and SW480 cells to 5-FU and Irinotecan upon to IFNγ stimulation. 200U IFNγ was administered 24 h before 48 h 100 μM drug treatment. The whiskers go down to the smallest value and up to the largest, the line in the box is plotted at the median. The box extends from the 25th to 75th percentiles computed by the GraphPad Prism (4 independent experiments, two-sided t test). (E, F) The ratio calculated by comparing the cell viability under drug alone or with IFNγ is clearly decreased for *CBX3* KO CRC cells but almost did not change in WT cells. $n = 3$ or 4 independent experiments, 2-sided t test. The data are presented as means ± SEM. (G) MC38 *Cbx3* KO cells is more sensible to IFNγ and increases more STAT1 expression after 24 h IFNγ stimulation compared to WT control. (H) *Cbx3* KO increased the sensitivity of MC38 cells to the treatment combined 5-FU and IFNγ. The whiskers go down to the smallest value and up to the largest, the line in the box is plotted at the median, the box extends from the 25th to 75th percentiles computed by the GraphPad Prism (6 independent experiments, two-sided t test). (I) Representative images of the tumors from 6 treatment groups at D22. (J) Tumor weights at D22 of 6 different treatment groups. The tumors from *Cbx3*KO treated with 5-FU combined IFNγ are significantly lighter than the tumors from WT group following the same treatment. The whiskers go down to the smallest value and up to the largest, the line in the box is plotted at the median, the box extends from the 25th to 75th percentiles computed by the GraphPad Prism ($n = 6$ or 7, two-sided t test). (K) Left: MC38*Cbx3* KO tumors significantly more sensitize to 5-FU + IFNγ treatment compared to WT tumors. Middle: *Cbx3* KO induce a tendency but do not significantly sensitize the MC38 to 5-FU treatment. Right: Compare to WT tumors which do not present a significant difference between 5-FU and 5-FU + IFNγ treatment, *Cbx3*KO tumors exhibit a significant difference between 5-FU and 5-FU + IFNγ treatment. Two-way ANOVA was applied for statistical analysis, $n = 5$ to 8 mice. The data are presented as means ± SEM. Source data are available online for this figure.

negative cells or by an LSR flow cytometer (BD) as Propidium Iodide (PI) negative cells.

## MC38 syngeneic mouse tumor models

C57BL/6 male mice (7 weeks) were from Janvier-Labs. Animal studies were approved by the ethical committee of Paris Descartes University (authorization number DAP 23-036). The mice were housed and elevated as previously described. After 1 week of settling down, a total of $0.5 \times 10^6$ MC38 WT and *Cbx3* KO cells were injected subcutaneously into C57BL/6 mice at their fourth inguinal nipple. After 1 week, mice started treatment, respectively, three times per week with DPBS (Control), 25 mg/kg Fluorouracil (5-FU, F0250000, Sigma-Aldrich) and 25 mg/kg 5-FU combined with 5000 U/mouse of recombinant marine IFNγ (315-05, Pepro Tech). For the mice treated with 5-FU and IFNγ, the first injection of IFNγ is 6 h before the 5-FU injection and the following injections are done at the same time. Tumor size was measured three times per week in two dimensions, the longest tumor dimension (L, length) and the value at right angles to it (W, width), with a caliper. Tumor volume was calculated as an ellipse (tumor volume = (width$^2$ * length)/2). After 16 days treatment, the mice were sacrificed and the tumors were separated and weight.

## Statistical analysis

For biological experiments, paired, unpaired, or multiple two-sided t-test and two-way ANOVA were performed with the GraphPad Prism based on at least three independent experiments or mice samples according to different experiment conditions. For syngeneic mouse tumor model, treatment groups were assigned in a randomized fashion but not blinded. For clinical analysis, sample size of the original TCGA COAD transcriptome cohort was taken in account with exclusion of samples comporting missing data which represent less than 0.3 percent of the total cohort. No randomization was performed during the observational studies. The two-sided t-test was applied for clinical co-expression analysis and the log-rank test of the Cox proportional Hazard model was applied for overall survival analysis. For all analysis, *P*-value inferior to 0.05 was considered as significant.

**The paper explained**

**Problem**
IFNγ is crucial for gut homeostasis and its dysregulation is linked to diverse colon pathologies, such as colitis and colorectal cancer (CRC). Given the important role of IFNγ/STAT1/PD-L1 axis in innate and adaptive immune responses, targeting the IFNγ signaling pathway could improve treatment efficacy for UC or CRC treatment.

**Results**
CBX3 was identified as an antagonist of the IFNγ signaling cascade in the colon epithelium via repression of STAT1 and CD274 transcription. Upon IFNγ stimulation, CBX3 binding to STAT1 and CD274 promoters was decreased, concomitant with their increased gene expression. CBX3 depletion led to a strong increase of STAT1 and PD-L1 expression under IFNγ stimulation, suggesting a role for CBX3 in the control of IFNγ-stimulated immune gene activation. Accordingly, CBX3 deletion resulted in chronic mouse colon inflammation, accompanied by upregulated STAT1 and CD274 expression. In particular, CBX3 deletion heightened CRC cells' sensitivity to IFNγ, which ultimately enhanced their chemosensitivity both in vitro and in vivo.

**Impact**
Our work identifies an interplay between CBX3 and IFNγ signaling that leads to regulation of the immune response in colon epithelium and of chemo-resistance of colorectal cancer. CBX3 appears as a potential target to enhance IFNγ-related therapeutic efficacy in UC and in colorectal cancer.

## Data availability

This study includes no data deposited in external repositories.

The source data of this paper are collected in the following database record: biostudies:S-SCDT-10_1038-S44321-024-00066-6.

## Peer review information

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

## Acknowledgements

HistIM Core Facility "Histology, Immunostaining, laser Microdissection", Institut Cochin, Inserm U1016, CNRS UMR8104, Université Paris Cité. Platform of LEAT, histology, Cytometry, SFR Necker Inserm US24 - CNRS UAR 3633. This work has been supported by the « Agence National de la Recherche» (ANR) (The project number: EPI-CURE, R16154KK) and INSERM (Institut National de la Santé et de la Recherche Médicale, annuel donation). Yao Xiang was supported by a fellowship from the China Scholarship Council (CSC) (n° 202008440433). Jorge Mata-Garrido was supported by ANR EPI-CURE, R16154KK. Yuanji Fu are supported by the CSC (n°202306380085). EP, CS, have received the support of the LabEx DCBIOL (ANR-10-IDEX-0001-02 PSL; INCa-DGOS-Inserm-12554, and Center of Clinical Investigation (CIC IGR- Curie 1428).

## Author contributions

**Yao Xiang**: Data curation; Investigation; Writing—review and editing. **Jorge Mata-Garrido**: Data curation; Investigation; Methodology. **Yuanji Fu**: Investigation; Writing—review and editing. **Christophe Desterke**: Data curation; Investigation; Writing—review and editing. **Eric Batsché**: Investigation; Methodology. **Ahmed Hamaï**: Writing—review and editing; provided scientific suggestions for the project. **Christine Sedlik**: Investigation; Methodology. **Youssouf Sereme**: Investigation. **David Skurnik**: Writing—review and editing; provided scientific suggestions for the project. **Abdelali Jalil**: Investigation; Visualization. **Rachel Onifarasoaniaina**: Investigation. **Eric Frapy**: Investigation. **Jean-Christophe Beche**: Investigation. **Razack Alao**: Investigation. **Eliane Piaggio**: Methodology; provided scientific suggestions for the project. **Laurence Arbibe**: Funding acquisition; Project administration; Writing—review and editing; provided scientific suggestions for the project. **Yunhua Chang**: Conceptualization; Data curation; Supervision; Funding acquisition; Investigation; Methodology; Writing—original draft; Project administration.

Source data underlying figure panels in this paper may have individual authorship assigned. Where available, figure panel/source data authorship is listed in the following database record: biostudies:S-SCDT-10_1038-S44321-024-00066-6.

## Disclosure and competing interests statement

The authors declare no competing interests.

# Expanded View Figures

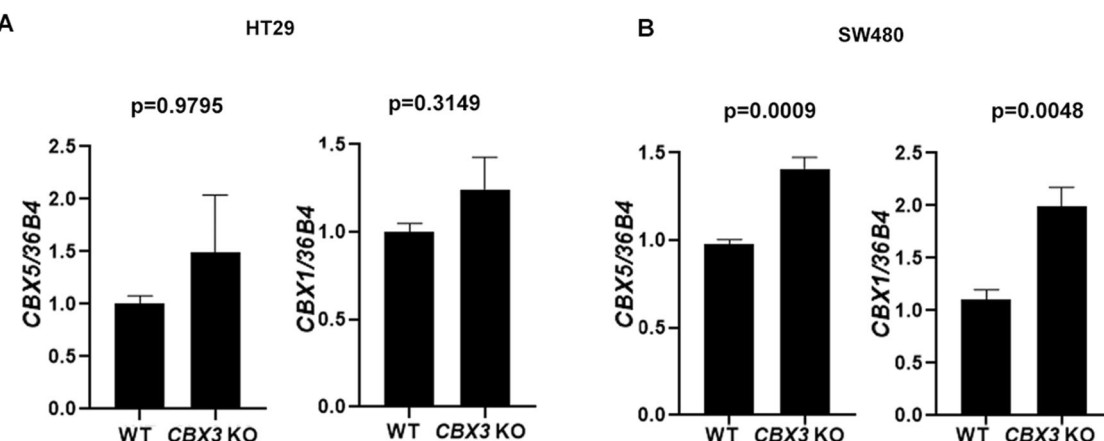

**Figure EV1. Increased *CBX5* and *CBX1* mRNA levels were found in SW480 *CBX3* KO cells but not in HT29 *CBX3* KO cells.**

(**A**) *CBX5* and *CBX1* mRNA levels are not significantly increased in HT29 CRISPR/Cas9 *CBX3* KO cells. (**B**) *CBX5* and *CBX1* mRNA levels are significantly increased in SW480 CRISPR/Cas9 *CBX3* KO cells. The error bar represented SEM. (4 independent experiments, two-sided t test).

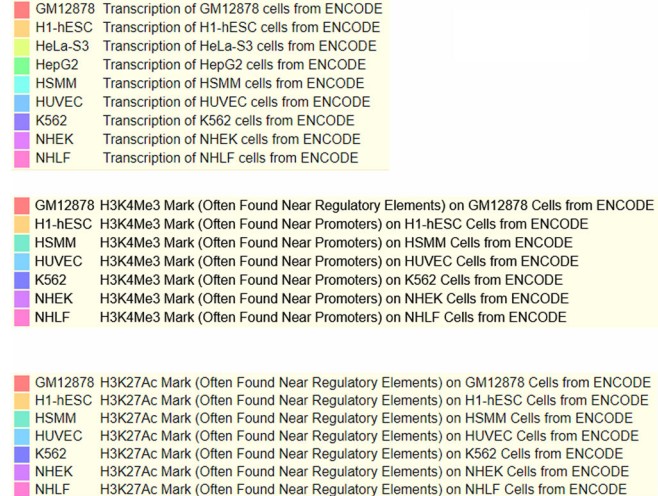

**Figure EV2.** The illustration of subtracks for transcription, H3K4Me3, and H3K27Ac Mark analysis from ENCODE.

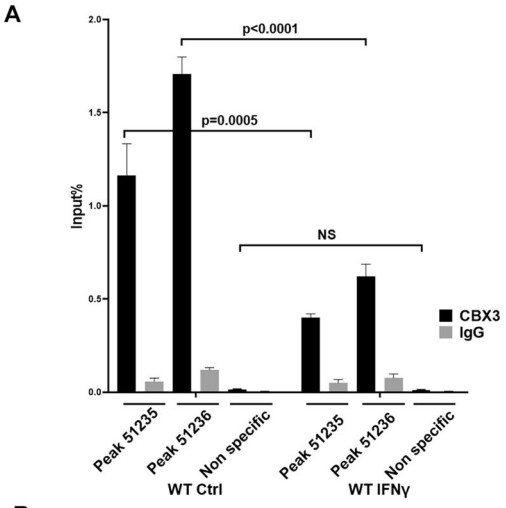

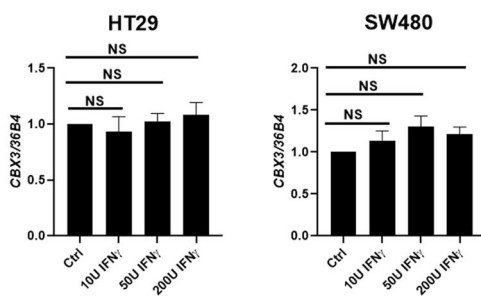

**Figure EV3.   ChIP-qPCR with specific primer compared to which with non-specific primers and *CBX3* mRNA level after 24h IFNγ stimulation.**

(A) Q-PCR with specific primers to Peak 51235 and 51236 regions compared to Q-PCR with primer sets aside from the indicated CBX3 binding domain of *STAT1* gene. The error bar represented SEM (3 or 4 independent experiments, two-way ANOVA test). (B) 24 h IFNγ stimulation did not modified CBX3 expression in HT29 and SW480 cells at the mRNA level. The error bar represented SEM (4 independent experiments, one-way ANOVA test).

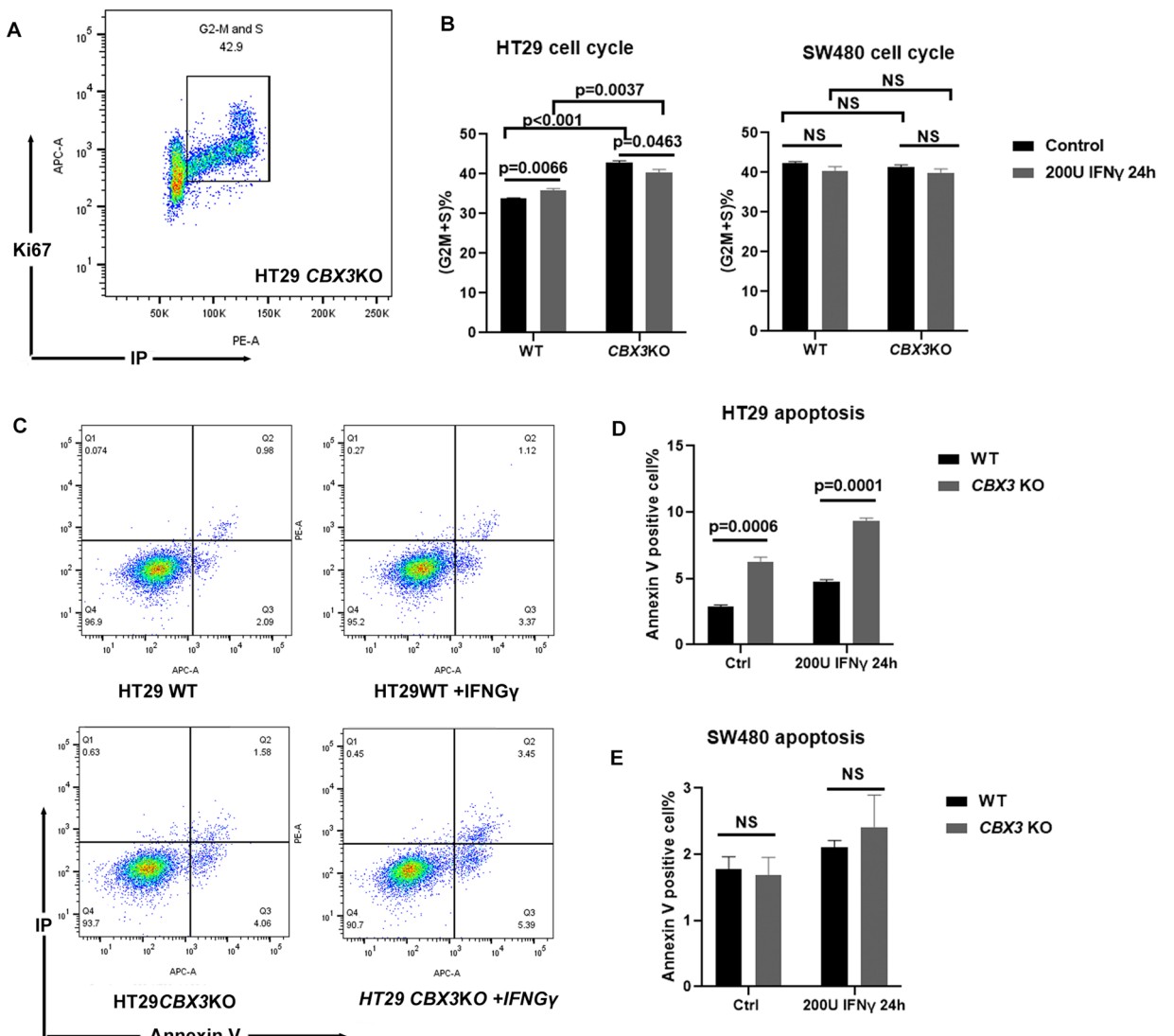

**Figure EV4.   The effect of CBX3 deletion and IFNγ stimulation on cell survival of HT29 and SW480 cells.**

(**A, B**) Flow Cytometry with anti-Ki67 revealed that *CBX3* KO led to higher cycling cell in HT29 cells but 72 h of IFNγ incubation reversed this effect. These effects are not observed in SW480 cells. The error bar represented SEM (4 independent experiments, two-sided t test). (**C–E**) *CBX3* deletion makes HT29 but not SW480 cells more sensitive to IFNγ induced apoptosis. The error bar represented SEM (3 independent experiments, two-sided t test).

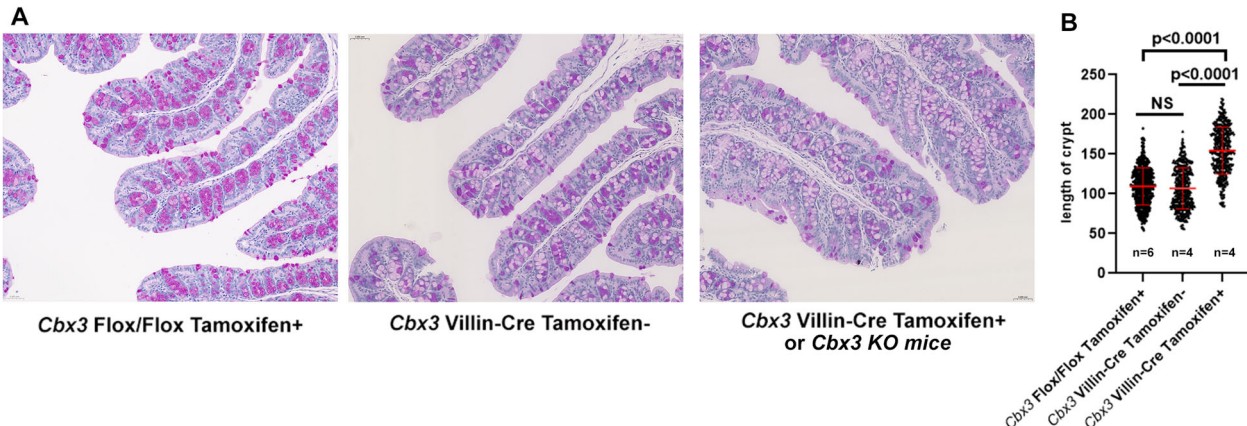

**Figure EV5. Additional controls using *Cbx3*Flox/Flox mice treated with Tamoxifen.**

(A) Crypt morphology and goblet cells density of *Cbx3*Flox/Flox mice treated with Tamoxifen show no colon inflammation criteria. (B) Crypt length exhibited no difference among *Cbx3* Villin-Cre mice treated with, without Tamoxifen and *Cbx3*Flox/Flox mice treated with Tamoxifen ($n = 4$ or 5, two-sided t test). The data are presented as means ± SD.

