## [Peer Review File · EMBO Molecular Medicine]

CBX3 antagonizes IFN γ /STAT1/PD-L1 axis to modulate colon inflammation and CRC chemosensitivity

Yao Xiang, Jorge Mata Garrido, Yuaji Fu, Christophe Desterke, Eric Batsché, Ahmed Hamaï, Christine Sedlik, Youssouf Sereme, David Skurnik, Abdelali Jalil, Rachel Onifarasoaniaina, Eric Frapy, Jean-Christophe Beche, Razack Alao, Eliane Piaggio, Laurence Arbibe, and Yunhua Chang-Marchand

Corresponding author: Yunhua Chang-Marchand (yunhua.chang-marchand@inserm.fr)

Review Timeline:

Submission Date:	26th Apr 23
Editorial Decision:	23rd May 23
Revision Received:	7th Feb 24
Editorial Decision:	22nd Feb 24
Revision Received:	14th Mar 24
Editorial Decision:	20th Mar 24
Revision Received:	27th Mar 24
Accepted:	3rd Apr 24

Editor: Lise Roth

Transaction Report:

23rd May 2023

Dear Dr. CHANG-MARCHAND,

Thank you for the submission of your manuscript to EMBO Molecular Medicine. We have now received feedback from the three referees who agreed to evaluate your work.

As you will see from the enclosed reports, the referees acknowledge the potential interest of the findings, however, they also raise several major concerns on the study, including the lack of in vivo validation of the in vitro data. Given the nature of the referees' concerns and the amount of time and work that would be required to address them, and considering that at EMBO Press we encourage one round of revisions only in a reasonable time frame, I am afraid I see little choice but to return the manuscript to you at this point with the decision that we cannot offer to publish it.

Given the potential interest of the findings, we would, however, have no objection to consider a new manuscript on the same topic if at some time in the near future you obtained data that would considerably strengthen the message of the study and address the referees concerns in full. To be completely clear, however, I would like to stress that if you were to send a new manuscript this would be treated as a new submission rather than a revision and would be reviewed afresh, in particular with respect to the literature and the novelty of your findings at the time of resubmission. If you decide to follow this route, please make sure you nevertheless upload a letter of response to the referees' comments.

At this stage of analysis, though, I am sorry to have to disappoint you. I nevertheless hope that the referee comments will be helpful in your continued work in this area and I thank you for considering EMBO Molecular Medicine.

Yours sincerely,

Lise Roth

***** Reviewer's comments *****

Referee #1 (Remarks for Author):

Xiang et al., identified CBX3 as an important negative regulator of IFN γ signaling in the colonic epithelium. Using RNAseq and ATACseq, as well as a variety of ex vivo and in vitro validations they show that loss of CBX3 increases the expression of PDL1 and STAT1 and significantly enhances the response of cancer cells to chemotherapy. As such, it could prove a promising therapeutic target, with potential impact also in immunotherapy treatment. The results are thus of high interest, the research is meticulous, and the manuscript is well written and clear.

- My main concern is the validation of the in vitro data in an in vivo setting. More specifically, how does CBX3 and the phenotype of CBX3 KO mice affect colitis and especially carcinogenesis in the intestine (e.g. using the DSS model, AOM/DSS model, orthotopic transplantations or combination with a genetic model)? Not all models are necessary, but a connection between CBX3 and colon cancer in vivo would better support the conclusions of the study. How does it affect chemotherapy treatment in vivo? These data are important to support the correlation analysis in the human data and the potential therapeutic value of targeting CBX3 in CRC.

- Better characterization of the inflammatory phenotype of CBX3 KO mice is needed, for example inflammatory cell infiltration using FACS to quantify both innate and adaptive immune subpopulations. Is the same phenotype seen also in the small intestine? If not, why could this be? Also, the authors should clarify what the WT controls are to be able to better interpret the data. Are they Cbx3f/f mice with tamoxifen? If not, this control should be shown to exclude any tamoxifen-associated changes especially in the early time-point.

- To better support role of CBX3 in the IFN γ /STAT1/PD-L1 axis, the effect on PD-L1 expression after inhibition of STAT1 in IFN γ -treated CBX3 cells should be shown.

Minor remarks:

- Quantification of Figure 1C would be helpful.
- In Figure 2A, "Around" should be omitted, since the exact number of genes is provided.
- In Line 259, it should be "Cbx3 KO mice"
- In Figure 3B, it should be "months"

- When referring to the PDL1 gene "Cd274" should be used.
- In Figure 4D, increased phosphorylation of STAT1 is not obvious in Figure 4D-middle panel. A more representative image should be shown or if not possible, the discrepancy should be discussed.

Referee #2 (Comments on Novelty/Model System for Author):

Although the basic findings may be interesting, however, only very few *in vivo* confirmations have been carried out, and experimental validation of major findings is missing. Please see below for further comments.

Referee #2 (Remarks for Author):

In this manuscript, Xiang Y et al describe an interplay between CBX3 and IFN-gamma, leading to the modified expression of STAT1 and PD-L1. They also provide data that CBX3 deletion results in the enhanced sensitivity of colorectal cancer (CRC) cells to IFN-gamma and to chemotherapeutic agents. Although the basic findings may be interesting, however, the manuscript suffers from several weaknesses. Some parts of the work are already known: i) the authors previously proved the effect of CBX3 deletion on colonic inflammation (Jorge Mata-Garrido et al, *Nat Commun*, 2022), ii) survival analysis has already been done for CBX3 (Xu H, *Front Med*, 2022). In addition, critical experiments have only been carried out *in silico* and *in vitro* with CRC cell lines, producing sometimes contradicting results.

Major comments:

1. Whereas mouse organoids respond to IFN-gamma, human CRC cells are unresponsive to this molecule. What can be the reason? The difference in 2D-3D models, species difference, or comparing normal colon and CRC cells? The authors should experimentally target this question (e.g. using human normal colon organoids, normal human colon cells etc).
2. Figure 4-5 show the lack of effect of IFN-gamma in CRC cell lines. Interestingly, IFN-gamma significantly decreased CBX3 chromatin association to the target genes. These two results seem to be contradictory. Thus, the authors should provide experimental data why STAT1 and PD-L1 are not regulated by IFN-gamma, even when the binding of CBX3 is inhibited to their promoters.
3. According to the title, CBX3 modulates colonic inflammation. Unfortunately, the mechanism is not clarified in the manuscript and only Fig1 contains some superficial analysis of this phenotype. Since one of the major messages of the manuscript is the modified PD-L1 level when CBX3 lacks, the functional importance of the IFN-gamma/CBX3/PD-L1 axis should be *in vivo* tested (e.g. by using blocking antibodies).
4. The authors conclude that the lack of CBX3 sensitizes CRC cells to IFN-gamma, leading to an increased chemosensitivity. However, data supporting this main finding derive purely from *in vitro* experiments using two CRC cell lines. Since these results are critical points in the manuscript and they represent the major novel findings, the authors should provide evidence from *in vivo* models as well (e.g. inducing colorectal tumors in wild type and knock-out mice and treating them with the drugs). Alternatively, human patient-derived organoids should be used that are a better model compared to 2D cell lines.
5. How does the increased PD-L1 expression modify chemosensitivity when CBX3 lacks? The interesting CBX3/PD-L1 loop should be experimentally addressed. At present the functional significance of PD-L1 in either inflammation or chemosensitivity is unclear in this manuscript.

Minor comments:

1. There are several spelling mistakes in the figures and text (e.g. line 694: "to mark the nuclear" should be nucleus; Fig1D: "crypte" should be crypt; Fig 2E,F: "chang" should be change; Fig 3B: "12 monthes" should be months etc).
2. Line 282-284: why is the compensatory increase in the mRNA levels of CBX5 and CBX1 important? Please provide some further explanation.
3. Fig 8: although figure legend and text say cell proliferation, but surviving cells are illustrated. Cell proliferation or apoptosis result in the differential cell survival?
4. Fig 1D shows difference in crypt length. However, this is not visible on the representative images in Fig 1B and C.
5. Fig 1C,G: Could the authors provide quantification of their results?
6. Fig 4D: In contrast to the manuscript text, the pSTAT1 signals on immunoblots from CBX3KO samples are not convincing.
7. Fig 7D-E: could you please give some more details about the tables?
8. Fig 7C: How was the cut-off value determined? The authors should provide some more details in the Materials section.
9. Fig. 8C-D: Could the authors show the statistical comparison of WT samples with/without IFN-gamma and CBX3KO samples with/without IFN-gamma? I think these comparisons could be relevant as well.
10. Fig 9: this figure is difficult to understand. Since IFN-gamma had no effect on STAT1/PD-L1, I think Fig 9B is not as simple as it is illustrated. In addition, no link between elevated PD-L1 and colonic inflammation has been proven in the manuscript. Similarly, the link between immune response and CRC chemosensitivity *in vitro* is unclear.

Referee #3 (Comments on Novelty/Model System for Author):

The authors utilise mouse model and colonic organoids in Figures 1 & 2. However, for the duration of the manuscript, they use

two CRC cell lines to substantiate their claims. I believe they should leverage their mouse model and organoid system to confirm and elevate the impact of their paper.

Referee #3 (Remarks for Author):

The manuscript from Xiang and colleagues investigates the role for the Cbx3 protein in regulating IFN- γ inflammatory signalling in the murine colon. Curiously, Cbx3 deletion in the colonic epithelium initiates an inflammatory response up to 12 months post-Cbx3 deletion. The authors follow up this finding by interrogating publicly available RNAseq and ChIPseq datasets to curate potential direct Cbx3 target genes that might be responsible for mediating inflammation. The authors identify putative Cbx3 binding sites in Stat1 and PD-L1 and show increased Stat1 and PD-L1 expression following Cbx3 deletion. The authors support this relationship using colonic organoids from WT and Cbx3KO mice to show expression of Stat1 and PD-L1 increase in Cbx3KO organoids compared to WT following IFN- γ treatment. The authors use HT29 and SW480 CRC cell lines to investigate and extend the relationship of Cbx3 in CRC. Using ChIP qPCR, the authors show enrichment for Cbx3 binding defined sites on PD-L1 and STAT1. From here, the manuscript makes claims based on findings that are correlative to suggest Cbx3-low expressing CRC have better survival. Yet, this is at odds with previously published roles of Cbx3 being a tumour-promoter. Analysis of patient survival is not convincing and doesn't properly stratify CRC patients based on CRC subtype. Additionally, given the authors have a conditional Cbx3flx allele, they should take advantage of this system to investigate if deletion of Cbx3 is cancer promoting or protective. Either AOM-DSS or crossing to Apc mutant mice would easily answer this question. Finally, the authors co-treat HT29 and SW480 cells with standard-of-care chemotherapy and IFN- γ to imply Cbx3-deficiency re-sensitises CRC cells to chemotherapy. This part of the paper is superficial and lacks substantial characterisation and explanation. If the authors leveraged their Cbx3 mouse model, they could properly explore the mechanisms that Cbx3 sensitise cells to chemotherapy. Does Cbx3 prevent CD8 and myeloid infiltrate, and thereby minimise effective cancer cell killing via chemotherapy? Overall, the manuscript presents conflicting data with previous roles for Cbx3 and performs multiple assays in inadequate model systems. I have listed several major and minor points that should be addressed before the manuscript is considered any further.

1. Crypt length is not a suitable surrogate for epithelial hyperplasia. Cell proliferation should be measured via Ki67 or BrdU staining.
2. The authors show changes to CD4 and CD8+ T-cells. IFN- γ is a well-known inducer of macrophages, so if the role of Cbx3 is to repress co-activators of downstream IFN- γ signalling, changes to other immune cells might be evident i.e. macrophages and neutrophils. This should be examined via IHC/IF or flow-cytometry.
3. The integrity of membrane permeability is often compromised during inflammation in the gut. Consequently, the authors need to test for this (either FITC-dextran or ZO-1 quantification), which may underpin increased immune infiltrate.
4. You can't use ChIP-seq from a cancer cell line and integrate it with RNAseq from a non-cancer setting. The mutation profile, and therefore the epigenetic landscape of the two systems will be dramatically different. ChIPseq from non-CRC Cbx3KO epithelial cells needs to be performed.
5. Does Cbx3 affect STAT1 binding to IFN γ -activation site (GAS) elements?
6. Figure 4C shows WT and Cbx3KO CRC cells treated with IFN γ . Intriguingly, IFN γ treatment increases Cbx3 expression compared to vehicle treated cells. This finding is counterintuitive to the proposed mechanism - Cbx3 expression should be reduced following IFN γ , and thereby allow expression of target genes STAT1 and PD-L1. In fact, the authors claim IFN γ stimulation significantly decreased CBX3 chromatin association on STAT1 and PD-L1 sites, which could not be attributed to a decrease of CBX3 level since 24h IFN γ stimulation did not result in any significant change in the expression of CBX3 at both the mRNA and protein levels in HT29. This clearly is not the result shown in Figure 4C. This finding undermines premise of the manuscript. Additionally, the expression of Cbx3 appears to be mostly cytoplasmic, even membranous rather than nuclear. The authors need to provide plausible evidence to explain both of these findings.
7. Given Cbx3KO induces an inflammatory response, do Cbx3KO organoids have a better ability to form organoids? Are they bigger or can regenerate following passage quicker compared to WT?
8. Cbx3 is uniformly expressed throughout the colonic epithelium (Fig 1B, Fig 3D). As such, deletion of an epigenetic modifier in all cells may elicit changes in stem cell fate, cell differentiation and ability to adapt following challenge. The authors should examine their RNAseq and ChIPseq for changes to gene signatures associated with these processes.
9. The manuscript looks at colonic biology in vivo at the beginning of the manuscript. But from Figure 3 onward, 2D CRC cell lines are used as model systems to extend their findings in a cancer context. Given the authors have access to mice and are capable of growing colonic organoids, they have missed a clear opportunity to strengthen their findings and elevate the quality of the paper. The most pressing and obvious experiment - is loss of Cbx3 cancer promoting or protective - is not evaluated at any point in the manuscript. The authors need to address this using mouse models - either AOM/DSS model or in Apcfl or ApcMin mice. At the very least CRC organoids should be used to confirm the findings from 2D CRC cell lines.
10. The notion that HT29 cells are non-responsive to IFN- γ is at odds with the current framework of IFN- γ /STAT1 signalling. In fact, recently published work from Coehlo et al. 2023 (Cancer Cell) show STAT1 KO in HT29 is a key mediator of IFN- γ resistance, implying STAT1 is responsive to IFN- γ signalling in HT29 cells. Furthermore, the blot for pSTAT1 in WT and Cbx3KO SW480 cells does not show increased pSTAT1 in Cbx3KO cells compared to WT. The authors should either repeat these experiments using a higher concentration of IFN- γ .

As a service to authors, EMBO provides authors with the possibility to transfer a manuscript that one journal cannot offer to publish to another EMBO publication. The full manuscript and if applicable, reviewers reports are automatically sent to the receiving journal to allow for fast handling and a prompt decision on your manuscript. For more details of this service, and to transfer your manuscript to another EMBO title please click on Link Not Available

***** Reviewer's comments *****

We appreciate you and the reviewers for providing invaluable feedback, which has

We sincerely thank all the reviewers' help and gladly followed their useful suggestions, which ultimately enhanced and clarified our manuscript. We responded to their comments in blue and detailed how we changed the manuscript in green.

Referee #1 (Remarks for Author):

Xiang et al., identified CBX3 as an important negative regulator of IFN γ signaling in the colonic epithelium. Using RNAseq and ATACseq, as well as a variety of ex vivo and in vitro validations they show that loss of CBX3 increases the expression of PDL1 and STAT1 and significantly enhances the response of cancer cells to chemotherapy. As such, it could prove a promising therapeutic target, with potential impact also in immunotherapy treatment. The results are thus of high interest, the research is meticulous, and the manuscript is well written and clear.

- My main concern is the validation of the in vitro data in an in vivo setting. More specifically, how does CBX3 and the phenotype of CBX3 KO mice affect colitis and especially carcinogenesis in the intestine (e.g. using the DSS model, AOM/DSS model, orthotopic transplantations or combination with a genetic model)? Not all models are necessary, but a connection between CBX3 and colon cancer in vivo would better support the conclusions of the study. How does it affect chemotherapy treatment in vivo? These data are important to support the correlation analysis in the human data and the potential therapeutic value of targeting CBX3 in CRC.

We thank the reviewer for this useful suggestion. We applied our authorization for animal experiments on a Syngeneic Mouse Tumor. As our *Cbx3*KO mice have a C57BL/6J background, we established MC38 Syngeneic Mouse Tumor Models. We first knocked out CBX3 in MC38 cell line and then grafted MC38 cells (WT and KO) to C57BL/6 mice. The formed tumors with MC38WT or MC38KO cells were then treated by 5-FU or 5-FU with IFN γ for 16 days. Tumor volumes were quantified 3 times each week to compare the growth rate between WT and KO tumors under different treatments. We also weigh the tumor at the end of treatment. Mice in vivo results are consistent with human CRC cells in vitro experiments, all of them revealed that the deletion of CBX3 increased CRC chemosensitivity, particularly under the IFN γ stimulation. All these data were added in **Figure 9G-9K** of the manuscript.

- Better characterization of the inflammatory phenotype of CBX3 KO mice is needed, for example inflammatory cell infiltration using FACS to quantify both innate and adaptive immune subpopulations.

We agree with the reviewer's opinion and tried to better characterize the colon inflammatory phenotype of *Cbx3* KO mice by histo-immunochemistry to check macrophage and neutrophil markers for innate immunity. Combined with our previous CD4 and CD8 T cells markers, this analysis also give us a global and directly visible inflammatory cell infiltration situation in the colon epithelium.

At the same time, we also quantified other inflammation phenotypes, as the goblet cell ratio as well as Ki67 and Brdu makers for colon epithelial hyperplasia linked to inflammation-induced regeneration. We checked also RNAseq analysis and revealed the proliferation related pathways, as G2M checkpoint, Mitotic spindle and E2F targets, are all enriched. These results indicate epithelial hyperplasia, which correspond to the increased inflammation and increased crypt length in KO epithelium.

All these data constitute two figures (Figure 1 and Figure 2) in the revised manuscript which permits better characterization of the inflammation phenotype.

Is the same phenotype seen also in the small intestine? If not, why could this be?

No. There exists no inflammation in the small intestine. Very interestingly, in contrast to colon epithelium, the immune response level seems to be downregulated in *Cbx3* KO intestine epithelium (see the figure showing GESA analysis). As *Cbx3* is an epigenetic factor, we supposed that CBX3 plays an important role in maintaining the immune homeostasis in the intestine and colon epithelium. According to the different microenvironments in the intestine and colon, the absence of CBX3 imbalances the immune response (or upregulated or downregulated). This is an interesting project and is on the way for this moment. However, it is a complete investigation in its own right and beyond the scope of this paper.

Figure for reviewers removed

Also, the authors should clarify what the WT controls are to be able to better interpret the data. Are they *Cbx3*^{f/f} mice with tamoxifen? If not, this control should be shown to exclude any tamoxifen-associated changes especially in the early time-point.

We accepted the reviewer's suggestion and clarified our mice model in the material and methods. Moreover, to exclude tamoxifen-associated inflammation, *Cbx3*^{Flox/Flox} mice that do not express Cre recombinase were identically treated with tamoxifen as an additional control. We then checked different colon inflammation criteria, such as crypt morphology, crypt length, and goblet cells in Tamoxifen-treated *Cbx3*^{Flox/Flox} mice. No inflammation was induced in Tamoxifen-treated *Cbx3*^{Flox/Flox} mice. We have added this point as a Supplemental Figure 1 and also modified the materials and methods in the text.

Briefly, Tamoxifen (0.5mg/mouse) diluted in 20% clinOleic acid was administrated by oral gavage, at 3 doses every 5 days in *Cbx3*^{Flox/Flox};Tg (Villin- CreERT2) mice, noted as *Cbx3* Villin-Cre or *Cbx3* KO mice in the later test. Control *Cbx3* Villin-Cre mice received 20% clinOleic acid alone by oral gavage. Additional controls using *Cbx3*^{Flox/Flox} mice that do not express the Cre recombinase were identically treated with Tamoxifen (**Figure S1**).

- To better support role of CBX3 in the IFN γ /STAT1/PD-L1 axis, the effect on PD-L1 expression after inhibition of STAT1 in IFN γ -treated CBX3 cells should be shown.

We thank the reviewer's suggestion and we have tried to infect HT29 *CBX3*KO and WT cells with a shRNA against *STAT1* coded in lentivirus (SIGMA). Unfortunately, we can't get a stable cell line after shRNA *STAT1* transduction in the *CBX3*KO cells. After a week of puromycin selection, the shRNA-transduced *CBX3*KO cells were in a very bad state and couldn't grow correctly to get a stable cell line. We supposed that's because the absence of CBX3 combined with the decrease of STAT1 makes the cells hard to live.

However, we could get a stable shRNA *STAT1* HT29 cell line and did the IFN γ stimulation as the reviewer asked. The Q-PCR results revealed that knocking down *STAT1* decreased *PD-L1* expression in both basal conditions and under IFN γ stimulation, which confirmed the existence of axis IFN γ /SATAT1/PD-L1 in CRC cells.

We have added these results in the paper as **Figure 6D**

Minor remarks:

- Quantification of Figure 1C would be helpful.

According to the suggestion of the reviewer, we have quantified goblet cells and added the results in new **Figure 2A**.

- In Figure 2A, "Around" should be omitted, since the exact number of genes is provided.

- In Line 259, it should be "Cbx3 KO mice"

- In Figure 3B, it should be "months"

- When referring to the PDL1 gene "Cd274" should be used.

Thanks a lot for the reviewer to indicate all these details. We have changed all the points as indicated by the reviewer.

- In Figure 4D, increased phosphorylation of STAT1 is not obvious in Figure 4D-middle panel. A more representative image should be shown or if not possible, the discrepancy should be discussed. We agree with the reviewer's remark, that the pSTAT1 signal on immunoblots, particularly in SW480 CBX3KO cells is quite weak. Because we have tried many times and can't get a satisfied western blot image, we suppose this is because of the lower pSTAT1 level in SW480 cells which arrived at the sensitivity limit of western blot. We then applied cytometry analysis by using an anti-STAT1 Phospho (Tyr701) PE antibody (Biolegend, 666403) to confirm our hypothesis.

As shown in the following figures, the IFN-gamma stimulation increased significantly p-STAT1 in CBX3KO cells. However, this increase is more evident in HT29 KO cells than in SW480 KO cells. All these results confirmed why we can detect an evident increased pSTAT1 by western blot in HT29CBX3KO cells but not in SW480CBX3KO under IFN-gamma stimulation.

This result was added in the paper as **Figure 5E**.

Referee #2 (Comments on Novelty/Model System for Author):

Although the basic findings may be interesting, however, only very few *in vivo* confirmations have been carried out, and experimental validation of major findings is missing. Please see below for further comments.

We thank the reviewer for this useful suggestion. We applied our authorization for animal experiments on a Syngeneic Mouse Tumor. As our Cbx3KO mice have a C57BL/6J background, we established MC38 Syngeneic Mouse Tumor Models. We first knocked out CBX3 in MC38 cell line and then grafted MC38 cells (WT and KO) to C57BL/6 mice. The formed tumors with MC38WT or MC38KO cells were then treated by 5-FU or 5-FU with IFN γ for 16 days. Tumor volumes were quantified 3 times each week to compare the growth rate between WT and KO tumors under different treatments. We also weigh the tumor at the end of treatment. Mice *in vivo* results are consistent with human CRC cells *in vitro* experiments, all of them revealed that the deletion of CBX3 increased CRC chemosensitivity, particularly under the IFN γ stimulation. All these data were added in **Figure 9G-9K** of the manuscript.

Referee #2 (Remarks for Author):

In this manuscript, Xiang Y et al describe an interplay between CBX3 and IFN- γ , leading to the modified expression of STAT1 and PD-L1. They also provide data that CBX3 deletion results in the enhanced sensitivity of colorectal cancer (CRC) cells to IFN- γ and to chemotherapeutic agents. Although the basic findings may be interesting, however, the manuscript suffers from several weaknesses. Some parts of the work are already known: i) the authors previously proved the effect of CBX3 deletion on colonic inflammation (Jorge Mata-Garrido et al, Nat Commun, 2022), ii) survival analysis has already been done for CBX3 (Xu H, Front Med, 2022). In addition, critical experiments have only been carried out *in silico* and *in vitro* with CRC cell lines, producing sometimes contradicting results.

Major comments:

1. Whereas mouse organoids respond to IFN- γ , human CRC cells are unresponsive to this molecule. What can be the reason?

It's a very interesting question. We also asked this question to ourselves and supposed that it could be due to the different IFN γ sensitivity between normal epithelial cells and cancer cells.

The "cancer immunoediting" concept exemplifies how cancer regressed its sensitivity to IFN- γ during tumor progression. As explicated in the review of Alspach et al. 2019 (Interferon γ and Its Important Roles in Promoting and Inhibiting Spontaneous and Therapeutic Cancer Immunity." Cold Spring Harbor Perspectives in Biology 11 (3). <https://doi.org/10.1101/cshperspect.a028480>), the process known as "cancer immunoediting" involves three stages: (1) elimination (2) equilibrium and (3) escape. Tumor cells that transit through the two first stages are subjected to severe immune-mediated selection pressures that lead to the removal of highly immunogenic tumor cell clones that are sensitive to IFN- γ . The outgrowth of tumor cells that are less immunogenic, not sensitive to IFN- γ and therefore are more fit to survive in an immunocompetent host (escape phase). According to this concept, normal colon epithelial cells and CRC cells could exhibit different sensitivities to IFN- γ , as normal colon organoid and human CRC cells.

Furthermore, even among CRC cells, different genome backgrounds, different mutation states, different development stages, treated (for human) or non-treated (for mouse), and even different treatment strategies could all affect their sensitivity to IFN γ . So, it's also completely possible to find different sensitivities to IFN γ stimulation among different human and mouse CRC cell lines, as HT29, SW480 and MC38.

The difference in 2D-3D models, species difference, or comparing normal colon and CRC cells? The authors should experimentally target this question (e.g. using human normal colon organoids, normal human colon cells etc).

We completely agree with the reviewer, there exist probably different responses to IFN-gamma among normal colon, different human and mice CRC 2D or 3D models.

In our revised version, we also checked a mice CRC cell line MC38. The mice 2D (MC38 cell line) or 3D (colon organoid) models exhibit a similar response pattern as mice colon organoids. The difference between human and mouse models, as we explained in the first question, could be due to many elements, such as their different genome backgrounds, different mutation states, different development stages, treated (for human) or non-treated (for mouse), etc..

Anyway, when we knock out CBX3, either mice colon epithelial cells or human or mice CRC cells all become more sensible to IFN-gamma stimulation. All these in vivo, ex vivo, and in vitro data from mice or human models support the same conclusion. The difference in 2D-3D models, species difference, or comparing normal colon and CRC cells is quite an interesting question and there exist many possibilities to explain these differences, which is inherently an interesting project for future research but is a complete investigation in its own right and beyond the scope of this paper.

2. Figure 4-5 show the lack of effect of IFN-gamma in CRC cell lines. Interestingly, IFN-gamma significantly decreased CBX3 chromatin association to the target genes. These two results seem to be contradictory. Thus, the authors should provide experimental data why STAT1 and PD-L1 are not regulated by IFN-gamma, even when the binding of CBX3 is inhibited to their promoters.

Although the increase is much slighter compared to CBX3KO cells, IFN-gamma treatment caused a rise in the expression of STAT1 and PD-L1 proteins as well as p-STAT1 in WT CRC cells, as demonstrated by western blot or cytometry analysis. The bands of STAT1 in WT cells are not very evident because the difference between WT and KO is too much, we expose a very short time in the Western experiment to evite the saturation of STAT1 band in KO cells.

The reviewer's remark makes us realize that our expression needs to be more precise. Regarding reviewer's comments, we modified the text about the description of the results as follows:

Remarkably, IFN γ stimulation limitedly increased STAT1 expression in WT cells, but strongly increased its mRNA and protein expression levels in two different CBX3 KO cell lines

Western

P-STAT1 cytometry

PD-L1 cytometry

Furthermore, as shown in our CHIP-QPCR data, IFN γ stimulation could decrease around 50% the binding of CBX3 to *STAT1* and *CD274* promoter but not abolish 100% the binding. The 50% left binding permits to increase in a rational *STAT1* and *CD274* expression under IFN γ stimulation, as shown in WT cells; but not a dramatic *STAT1* and PD-L1 expression, as it happened in *CBX3* KO cells. We thus suppose that the strict controlling of IFN γ signaling activation is an important protection mechanism of the cells. Too high IFN γ signaling activation could induce uncontrolled consequences, as apoptosis, cell cycle arrest as well as chemosensitivity, as it's happened in HT29 cells (Figure S4 and Figure 9).

3. According to the title, CBX3 modulates colonic inflammation. Unfortunately, the mechanism is not clarified in the manuscript and only Fig1 contains some superficial analysis of this phenotype. Since one of the major messages of the manuscript is the modified PD-L1 level when CBX3 lacks, the functional importance of the IFN- γ /CBX3/PD-L1 axis should be *in vivo* tested (e.g. by using blocking antibodies).

5. How does the increased PD-L1 expression modify chemosensitivity when CBX3 lacks? The interesting CBX3/PD-L1 loop should be experimentally addressed. At present the functional significance of PD-L1 in either inflammation or chemosensitivity is unclear in this manuscript. It's really an important and interesting question.

As these two questions are closely related, I answer them together.

Firstly, we appreciated the reviewer's remark and tried to better characterize the colon inflammatory phenotype of *CBX3* KO mice. We have checked macrophage and neutrophil markers for innate immunity. Combined with our previous CD4+ and CD8+ T cells markers, this analysis could also give us a global and directly visible inflammatory cell infiltration situation in the colon epithelium.

At the same time, we quantified other inflammation phenotypes, as the goblet cell ratio as well as Ki67 and BrdU makers for colon epithelial hyperplasia linked to inflammation-induced regeneration. We checked also RNAseq analysis and revealed the proliferation related pathways, as G2M checkpoint, Mitotic spindle and E2F targets, are all enriched. These results indicate epithelial hyperplasia, which correspond to the increased inflammation and increased crypt length in KO epithelium.

All these data constitute two figures (Figure 1 and Figure 2) in the revised manuscript which permits better characterization of the inflammation phenotype.

About the functional importance of the IFN- γ /CBX3/PD-L1, we completely agree with the review and thanks a lot for this constructive suggestion. In the new version of the paper, we established MC38 Syngeneic Mouse Tumor Models to graft WT and *CBX3*KO tumors. *In vivo* results from syngeneic mouse tumor models are consistent with *in vitro* results from human CRC cells. The deletion of *CBX3* increased the chemosensitivity, particularly under the IFN γ stimulation. This experiment could partially answer the *in vivo* function importance of *Cbx3*/IFN γ axis.

Furthermore, as suggested by the reviewer and we discussed in the paper, the functional importance of IFN- γ /CBX3/PD-L1 certainly deserves further investigation. Nevertheless, the main goal of this work is to show the antagonism of *CBX3* to IFN γ and thus mediate inflammation and

chemotherapy sensitivity. Another independent project to check their role in CRC cancer treatment is on the way, in which anti PD-L1 monoclonal antibody will be applied in MC38 Syngeneic Mouse Tumor Models grafted WT and CBX3KO tumors. In this independent investigation, we could answer completely how IFN-gamma/CBX3/PD-L1 axis affects CRC immunotherapy as well as how the modification of PD-L1 expression affected the efficacy of different therapies.

4. The authors conclude that the lack of CBX3 sensitizes CRC cells to IFN-gamma, leading to an increased chemosensitivity. However, data supporting this main finding derive purely from in vitro experiments using two CRC cell lines. Since these results are critical points in the manuscript and they represent the major novel findings, the authors should provide evidence from in vivo models as well (e.g. inducing colorectal tumors in wild type and knock-out mice and treating them with the drugs). Alternatively, human patient-derived organoids should be used that are a better model compared to 2D cell lines.

Thanks reviewer's remarkable comment. According to the reviewer's suggestion, we established MC38 Syngeneic Mouse Tumor Model which is grafted in C57BL/6 mice.

We first knocked out *Cbx3* in MC38 cell line and then grafted MC38 cells (WT and CBX3KO) to C57BL/6 mice. The formed tumors were then treated by 5-FU with or without IFN γ for 16 days. Tumor volumes were quantified 3 times each week to compare the growth rate between WT and CBX3KO tumors under different treatments. As shown in Figure 9, mice *in vivo* results are consistent with human CRC cells in vitro experiments, which revealed that the deletion of CBX3 increased CRC chemosensitivity, particularly under the IFN γ stimulation

Minor comments:

1. There are several spelling mistakes in the figures and text (e.g. line 694: "to mark the nuclear" should be nucleus; Fig1D: "crypte" should be crypt; Fig 2E,F: "chang" should be change; Fig 3B: "12 monthes" should be months etc).

Thanks a lot for the indication. We have modified these mistakes as indicated by the reviewer.

2. Line 282-284: why is the compensatory increase in the mRNA levels of CBX5 and CBX1 important? Please provide some further explanation.

Yes, we have firstly added some further explanation in the introduction about HP1 family.

The heterochromatin protein 1 (HP1) family are epigenetic regulators, which are readers of the H3K9me2/3 histone modifications and play an important role in the formation and maintenance of heterochromatin. HP1 family includes CBX1 (HP1 β), CBX5 (HP1 α) and CBX3 (HP1 γ) in mouse and human.

And then, the potential influence of this compensation among HP1 in the discussion:

Interestingly, deletion CBX3 in SW480 or HT29 cells is enough to induce dramatic STAT1 and CD274 expression under IFN γ stimulation, even though there exists a compensatory increase of CBX5 (HP1 α) mRNA and CBX1 (HP1 β) mRNA is identified in SW480 cells. CBX3 antagonist effect to IFN γ could not be compensated by other members of HP1 family.

3. Fig 8: although figure legend and text say cell proliferation, but surviving cells are illustrated. Cell proliferation or apoptosis result in the differential cell survival?

Thanks for reviewer's opinion. As reviewer said the different cell proliferation and apoptosis states result in different cell survival, thus we have checked the cell proliferation state by marking Ki67+PI (Figure S4 A-B) as well as apoptosis states by Annexin V (Figure S4 C-E).

Corresponding to the ancient Figure 8A (new Figure 9A-9B), KO CBX3 and IFN-gamma stimulation in SW480 didn't significantly change the cell proliferation (G2M and S phases cycling cells) and cell apoptosis (Annexin V positive cells), thus the final relative SW480 KO living cells number keep stable and comparable to WT cells under different conditions.

However, KO *CBX3* increased both cell proliferation and apoptosis in HT29 cells, but the increase of G2M and S phases cells ratio is much higher than apoptosis cells ratio, the HT29 *CBX3*KO cells thus exhibit higher living cell number. But IFN-gamma stimulation not only decreased G2M and S phases cell ratio but also **increased** the apoptosis ratio in HT29 *CBX3*KO cells. Together, less viable HT29 KO cells were detected compared to WT cells under IFN-gamma stimulation.

4. Fig 1D shows difference in crypt length. However, this is not visible on the representative images in Fig 1B and C.

For more easily understand the measurement, we have indicated directly the measurement with a red trait in the figure (See new Figure 2A)

5. Fig 1C,G: Could the authors provide quantification of their results?

According to the suggestion of the reviewer, we have added the goblet cell quantification (New Figure 2A).

6. Fig 4D: In contrast to the manuscript text, the pSTAT1 signals on immunoblots from *CBX3*KO samples are not convincing.

Thanks for reviewer's comments. We agree that the pSTAT1 signal on immunoblots, particularly in SW480 *CBX3*KO cells is quite weak. Because we have tried many times and can't get a satisfied western blot image, we suppose this is because of the lower pSTAT1 level in SW480 cells which arrived at the sensitivity limit of western blot. We then applied cytometry analysis by using an anti-STAT1 Phospho (Tyr701) PE antibody (Biolegend, 666403) to confirm our hypothesis.

As shown in the following figure, the IFN-gamma stimulation increased significantly p-STAT1 in *CBX3*KO cells. However, this increase is more evident in HT29 KO cells than in SW480 KO cells. All these results confirmed why we can detect an evident increased pSTAT1 by western blot in HT29*CBX3*KO cells but not in SW480*CBX3*KO under IFN-gamma stimulation.

This result was added in the paper as Figure 5E.

7. Fig 7D-E: could you please give some more details about the tables?

According to the remark of the reviewer, we have added more information to the figure legend of Figure 8E (ancient Figure 7D-E).

Time is represented in months. The patients' number associated with each group is showed in the under table. Tables under Kaplan Meier graph represent patients still at risk for the distinct times of follow (0,50,100,150 months) according the molecular group stratification.

8. Fig 7C: How was the cut-off value determined? The authors should provide some more details in the Materials section.

As indicated in the material and method, this cut-off is done by Logicile R automatically.

9. Fig. 8C-D: Could the authors show the statistical comparison of WT samples with/without IFN-gamma and CBX3KO samples with/without IFN-gamma? I think these comparisons could be relevant as well.

We accept the reviewer's suggestion and have added the statistical comparison in the new Figure 9C and 9D (ancient figure 8C and 8D).

10. Fig 9: this figure is difficult to understand. Since IFN-gamma had no effect on STAT1/PD-L1, I think Fig 9B is not as simple as it is illustrated. In addition, no link between elevated PD-L1 and colonic inflammation has been proven in the manuscript. Similarly, the link between immune response and CRC chemosensitivity in vitro is unclear.

In the new version, we have removed this figure as is not enough precise to present the manuscript, as suggested by the reviewer

Referee #3 (Comments on Novelty/Model System for Author):

The authors utilise the mouse model and colonic organoids in Figures 1 & 2. However, for the duration of the manuscript, they use two CRC cell lines to substantiate their claims. I believe they should leverage their mouse model and organoid system to confirm and elevate the impact of their paper. We agree with the reviewer on this point. According to the suggestion of the reviewer, we have applied the experimental authorization for the mice cancer model. As our CBX3KO mice are the C57BL/6J background, we thus established MC38 Syngeneic Mouse Tumor Model which is grafted in C57BL/6 mice.

We first knocked out CBX3 in MC38 cell line and then grafted MC38 cells (WT and KO) to C57BL/6 mice. The formed tumors with MC38WT or MC38KO cells were then treated by 5-FU with or without IFN γ for 16 days. Tumor volumes were quantified 3 times each week to compare the growth rate between WT and CBX3KO tumors under different treatments. We weigh also the tumor at the end of treatment. As shown in Figure 9, mice *in vivo* results are consistent with human CRC cells *in vitro* experiments, all of them revealed that the deletion of CBX3 increased CRC chemosensitivity, particularly under the IFN γ stimulation.

Referee #3 (Remarks for Author):

The manuscript from Xiang and colleagues investigates the role for the Cbx3 protein in regulating IFN- γ inflammatory signaling in the murine colon. Curiously, Cbx3 deletion in the colonic epithelium initiates an inflammatory response up to 12 months post-Cbx3 deletion. The authors follow up this finding by interrogating publicly available RNAseq and ChIPseq datasets to curate potential direct Cbx3 target genes that might be responsible for mediating inflammation. The authors identify putative Cbx3 binding sites in Stat1 and PD-L1 and show increased Stat1 and PD-L1 expression following Cbx3 deletion. The authors support this relationship using colonic organoids from WT and Cbx3KO mice to show expression Stat1 and PD-L1 increase in Cbx3KO organoids compared to WT following IFN- γ treatment. The authors use HT29 and SW480 CRC cell lines to investigate and extend the relationship of Cbx3 in CRC. Using ChIP qPCR, the authors show enrichment for Cbx3 binding defined sites on PD-L1 and STAT1. From here, the manuscript makes claims based on findings that are correlative to suggest Cbx3-low expressing CRC have better survival. Yet, this is at odds with previously published roles of Cbx3 being a tumour-promoter. Analysis of patient survival is not convincing and doesn't not properly stratify CRC patients based on CRC subtype. Additionally, given the authors have a conditional Cbx3flx allele, they should take advantage of this system to investigate if deletion of Cbx3 is cancer promoting or protective. Either AOM-DSS or crossing to Apc mutant mice would easily answer this question. Finally, the authors co-treat HT29 and SW480 cells with standard-of-care chemotherapy and IFN- γ to imply Cbx3-deficiency re-sensitises CRC cells to chemotherapy. This part of the paper is superficial and lacks substantial characterisation and explanation. If the authors leveraged their Cbx3 mouse model, they could properly explore the mechanisms that Cbx3 sensitise cells to chemotherapy. Does Cbx3 prevent CD8 and myeloid infiltrate, and thereby minimise effective cancer cell killing via chemotherapy? Overall, the manuscript presents conflicting data with previous roles for Cbx3 and performs multiple assays in inadequate model systems. I have listed several major and minor points that should be addressed before the manuscript is considered any further.

1. Crypt length is not a suitable surrogate for epithelial hyperplasia. Cell proliferation should be measured via Ki67 or BrdU staining.

We thank the reviewer for this suggestion. In this revised manuscript, we have checked Ki67 and BrdU staining in the colon epithelium of WT and KO mice. The Ki67-positive and BrdU-positive cells are significantly increased in CBX3KO mice colon epithelium. We also checked RNAseq analysis and revealed the proliferation-related pathways, as the G2M checkpoint, mitotic spindle, and E2F targets are all enriched. These results indicate epithelial hyperplasia, which is linked to inflammation-induced regeneration due to the increased inflammation and increased crypt length in KO epithelium.

These results were added in new **Figure 2C-2F**.

2. The authors show changes to CD4 and CD8+ T-cells. IFN- γ is a well-known inducer of macrophages, so if the role of Cbx3 is to repress co-activators of downstream IFN- γ signalling, changes to other immune cells might be evident ie. macrophages and neutrophils. This should be examined via IHC/IF or flow-cytometry.

We thank the reviewer's suggestion and checked macrophage and neutrophil markers for innate immunity by immunohistology. Combined with our previous CD4+ and CD8+ T cells markers, this analysis could also give us a global and directly visible inflammatory cell infiltration situation in the colon epithelium.

3. The integrity of membrane permeability is often compromised during inflammation in the gut. Consequently, the authors need to test for this (either FITC-dextran or ZO-1 quantification), which may underpin increased immune infiltrate.

We agree with the reviewer's opinion that the integrity of membrane permeability could underpin increased immune infiltration. We also thought about this possibility at the beginning of our work. However, the GSEA analysis based on our RNAseq data revealed that the apical junction pathway is enriched in the colon epithelium of Cbx3 KO mice (7 days after tamoxifen induction), which suggested increased immune infiltration induced by Cbx3 KO is not due to the modification of the integrity of membrane permeability.

4. You can't use ChIP-seq from a cancer cell line and integrate it with RNAseq from a non-cancer setting. The mutation profile, and therefore the epigenetic landscape of the two systems will be dramatically different. ChIPseq from non-CRC Cbx3KO epithelial cells needs to be performed.

We agree to the review, there indeed exist many differences between cancer and no cancer system. However, the objective of integrating ChIP-seq from HCT116 with RNAseq from mice colon epithelium is just to look for the potential CBX3 targeting genes whose promoters are bound to CBX3 and whose expressions are modified by Cbx3KO in our model, which could give us useful clues for further study. The Stat1 and PD-L1 that appeared from this analysis are just considered as potential targets of CBX3 at the beginning of this study. Then, different experimental models, such as mice colon epithelium, mice colon organoid or CRC cell lines from humans and mice were used to confirm this analysis results. As we could see in our study, even there probably exist many differences in different models, when we knock out CBX3, either mice normal colon epithelial cells or human or mice CRC cell lines all become more sensible to IFN-gamma and increased STAT1 and PD-L1 expression under IFN-gamma stimulation. These data suggest CBX3 antagonizes IFN-gamma via regulating STAT1 and PD-L1 transcription is a common molecular mechanism between normal epithelial cells and CRC cells.

We agree with the reviewer that ChIPseq from non-CRC Cbx3KO epithelial cells could exhibit many differences compared to ChIPseq from CRC cell. It will be perfect to compare these two ChIP-seq sets, which could provide many useful clues to identify new therapeutic targets for CRC treatment. Further ChIPseq experiments are under consideration.

5. Does Cbx3 affect STAT1 binding to IFN γ -activation site (GAS) elements?

This is a very interesting but quite complex question. Using the case of STAT1 as an example, the GAS element (marked in yellow in the attached figure) is located in one of the CBX3 binding peak 51236 located in STAT1 promotor. As shown in our ChIP-QPCR data, IFN γ stimulation could decrease around 50% of the binding of CBX3 to peak 51236 but not abolish 100% the binding. The 50% left binding of CBX3 permits to increase in a rational STAT1 expression under IFN γ stimulation, as shown in WT cells; but not a dramatic STAT1 expression, as it happened in *CBX3* KO cells. This effect suggests the existence of 50% CBX3 binding could limit enough pSTAT1 binding to GAS elements for inducing a dramatic STAT1 transcription. However, this conclusion needs much more experimental data to confirm. Unfortunately, we don't have enough data to answer this question at this moment.

Genomic chr2 (reverse strand):191020348-191020516

CGGGAAAGCC CCGCTGCGCC TGC GTGCTGC GCCTGCGTAC TGCTCCCGCA 191020467
 GGCTCTAACT TTCGCTTTTC GGTTTCCTCT CAATCCAGT CTTTCTCGCT 191020417
 CCCGGGCAGT **TTCCCCGAAA** GCACGTGGGG CGGCTCTCC GAGCACTCGC 191020367
 GTCCTCCCG CCGCGCGCG

6. Figure 4C shows WT and Cbx3KO CRC cells treated with IFN γ . Intriguingly, IFN γ treatment increases Cbx3 expression compared to vehicle treated cells. This finding is counterintuitive to the proposed mechanism Cbx3 expression should be reduced following IFN γ , and thereby allow expression of target genes STAT1 and PD-L1. In fact, the authors claim IFN γ stimulation significantly decreased CBX3 chromatin association on STAT1 and PD-L1 sites, which could not be attributed to a decrease of CBX3 level since 24h IFN γ stimulation did not result in any significant change in the expression of CBX3 at both the mRNA and protein levels in HT29. This clearly is not the result shown in Figure 4C. This finding undermines premise of the manuscript.

We understand the review's doubt and thank the reviewer indicated this point which we didn't pay attention to at the beginning. We suggest this result is just due to a manipulation problem. We then rechecked this result with two different techniques: 1) The Q-PCR results showed that 24h IFN γ treatment did not significantly increase *CBX3* RNA expression. 2) the western blot result with several different series samples did not show an increased expression of CBX3 after 24h IFN-g stimulation.

Additionally, the expression of Cbx3 appears to be mostly cytoplasmic, even membranous rather than nuclear. The authors need to provide plausible evidence to explain both of these findings.

According to our data, CBX3 is mostly expressed in the nucleus and especially around the nuclear membrane, either in mice colon epithelium or in CRC cells. As shown in the following figure for CRC cell line, the blue is DAPI which marks the nuclear and the red is CBX3 marker which is colocalized within the nucleus but more evident in the nuclear peripheral area.

SW480 cells

7. Given Cbx3KO induces an inflammatory response, do Cbx3KO organoids have a better ability to form organoids? Are they bigger or can regenerate following passage quicker compared to WT?

Thanks for reviewer's comment. According to the literature, the Inflammation of the intestine or colon organoid reduces the organoid size: 1). Characterization of Human Colon Organoids from Inflammatory

Bowel Disease Patients. *Front Cell Dev Biol.* 2020 Jun 4;8:363. doi: 10.3389/fcell.2020.00363. 2). FUT2-dependent fucosylation of HYOU1 protects intestinal stem cells against inflammatory injury by regulating unfolded protein response. *Redox Biol.* 2023 Apr;60:102618. doi: 10.1016/j.redox.2023.102618.

From our preliminary data, Cbx3KO seems less easy to form colon organoids. Regarding the size of organoids, we measured the diameter of Cbx3KO and WT colon organoids by image J and then calculated their volume. Our results revealed that KO organoids (n = 21) exhibited a smaller size compared to WT organoids (n = 16), which corresponded to their higher inflammation response. More detailed analysis is on the way in another independent project to understand the different inflammation characteristics in colon and intestine organoids after deletion of CBX3.

8. Cbx3 is uniformly expressed throughout the colonic epithelium (Fig 1B, Fig 3D). As such, deletion of an epigenetic modifier in all cells may elicit changes in stem cell fate, cell differentiation and ability to adapt following challenge. The authors should examine their RNAseq and ChIPseq for changes to gene signatures associated with these processes.

As suggested by the reviewer, we have examined RNAseq data to check if there exist changes in cell differentiation, proliferation, and ability to adapt following challenges.

Most evidently, the proliferation pathway as G2M checkpoint, Mitotic spindle and E2F targets are increased are enriched in Cbx3KO colon crypt. All of these correspond to the increased inflammation and crypt length (Hyperplasia). We have added this information to **Figure 2F**. We also confirmed this observation by Ki67 and Brdu marking on the colon histological section, which is added in **Figure 2C-2E**.

9. The manuscript looks at colonic biology in vivo at the beginning of the manuscript. But from Figure 3 onward, 2D CRC cell lines are used as model systems to extend their findings in a cancer context. Given the authors have access to mice and are capable of growing colonic organoids, they have missed a clear opportunity to strengthen their findings and elevate the quality of the paper. The most pressing and obvious experiment - is loss of Cbx3 cancer promoting or protective - is not evaluated at any point in the manuscript. The authors need to address this using mouse models - either AOM/DSS model or in Apcfl or ApcMin mice. At the very least CRC organoids should be used to confirm the findings from 2D CRC cell lines.

We completely agree with the reviewer's opinion. According to the suggestion of reviewer, we have applied the authorization for animal experiments to realize the most pressing and obvious experiment to check how loss of CBX3 affects CRC development and treatment.

As our CBX3KO mice are the C57BL/6J background, we thus established MC38 Syngeneic Mouse Tumor Model which is grafted in C57BL/6 mice.

We first knocked out CBX3 in MC38 cell line and then grafted MC38 cells (WT and KO) to C57BL/6 mice. The formed tumors with MC38WT or MC38KO cells were then treated by 5-FU or 5-FU with IFN γ for 16 days. Tumor volumes were quantified 3 times each week to compare the growth rate between WT and KO tumors under different treatments. We weigh also the tumor at the end of treatment. As showed in **Figure 9H-9K**.

Mice in vivo results are consistent to human CRC cells in vitro experiments, all of them revealed that the deletion of CBX3 increased CRC chemosensitivity under the IFN γ stimulation

10. The notion that HT29 cells are non-responsive to IFN- γ is at odds with the current framework of IFN- γ /STAT1 signalling . In fact, recently published work from Coehlo et al. 2023 (Cancer Cell) show STAT1 KO in HT29 is a key mediator of IFN- γ resistance, implying STAT1 is responsive to IFN- γ signalling in HT29 cells.

We appreciate the reviewer's feedback. In fact, although the increase is much slighter compared to CBX3KO cells, IFN-gamma treatment caused a rise in the expression of STAT1 and PD-L1 proteins as well as p-STAT1 in WT CRC cells, as demonstrated by western blot or cytometry analysis. The bands of STAT1 in WT cells are not very evident because the difference between WT and KO is too much, we expose a very short time in the Western experiment to evite the saturation of STAT1 band in KO cells. The reviewer's remark makes us realize that our expression needs to be more precise. Regarding reviewer's comments, we modified the text about the description of the results as follows:

Remarkably, IFN γ stimulation limitedly increased STAT1 expression in WT cells, but strongly increased its mRNA and protein expression levels in two different CBX3 KO cell lines

Western

P-STAT1 cytometry

Furthermore, the blot for pSTAT1 in WT and CBx3KO SW480 cells does not show increased pSTAT1 in Cbx3KO cells compared to WT. The authors should either repeat these experiments using a higher concentration of IFN- γ .

We agree with the reviewer's remark, the pSTAT1 signals on immunoblots, particularly in SW480 CBX3KO cells is quite weak. Because we have tried many times and can't get a satisfied western blot image, we suppose this is because of the lower pSTAT1 level in SW480 cells which arrived at the sensitivity limit of western blot. We then applied cytometry analysis by using an anti-STAT1 Phospho (Tyr701) PE antibody (Biolegend, 666403) to confirm our hypothesis.

As shown in the following figures, the IFN-gamma stimulation increased significantly p-STAT1 in CBX3KO cells. However, this increase is more evident in HT296 KO cells than in SW480 KO cells. That's why we can detect an evident increased pSTAT1 by western blot in HT29CBX3KO cells but not in SW480CBX3KO under IFN-gamma stimulation.

Our objective is to show that KO CBX3 increased the sensitivity of CRC cells to IFN- γ . In WT CRC cells as shown in western blot or cytometry analysis, IFN-gamma induced an increased expression of STAT1 and PD-L1 as well as p-STAT1 but these increases are much slighter compared to CBX3KO cell lines. The concentration used for our experiment is suitable to reveal this difference.

22nd Feb 2024

Dear Dr. Chang-Marchand,

Thank you for submitting your revised manuscript. We have now received the reports from referees #1 and #2 who who re-reviewed your manuscript. As you will see below, they are satisfied with the revisions. Referee #3 was unfortunately not able to evaluate the revised manuscript, however referees #1 and #2 further assessed your responses to this referee's concerns and stated:

Referee #1:

"The authors performed additional experiments to support their conclusions and clarify their results, based on the comments of referee #3. Most notably, they provide an in vivo validation of their results using a xenograft model. Since this experiment does not utilize the CBX3KO mice, it cannot provide a connection with the initially described inflammation and hyperplasia. Nevertheless, it provides in vivo evidence that supports the in vitro mechanistic data.

There are two comments from the reviewer that were not answered with new experiments.

In comment 3, the authors did not perform experiments on epithelial permeability but show that their data do not point to any difference. As such, these experiments are not expected to affect the conclusions of the paper.

In comment 4, the authors did not perform new ChIP experiments, as the reviewer suggested. I am not an expert in ChIPseq, but I understand the problem of integrating data from cells in different mutation states. Nevertheless, the scope of this analysis is limited and the results are experimentally validated. If the authors cannot perform these experiments, they should acknowledge the limitation of their analysis in the manuscript.

To my opinion, the revised manuscript is suitable for publication."

Referee #2:

"I think the authors responded to reviewer#3's questions successfully, they carried out the necessary experiments and included novel data into the manuscript. Some critical issues raised by more reviewers (e.g. confirmation by in vivo experiments, further characterization of the intestinal inflammation, problems with the detection of pSTAT1) have also been solved by the authors."

I will therefore be able to accept your manuscript once the following editorial points will be addressed:

1/ Manuscript text:

- The order of first/last name in the manuscript is different in the manuscript from the submission system - Elaine Piaggio vs. Piaggio Elaine in the manuscript. The correct order should be first name/last name.
- Please remove the green text, and only keep in track changes mode any new modification.
- We can accommodate a maximum of 5 keywords, please adjust accordingly.
- We do not have list of abbreviations. Please remove it from the manuscript text, and define each abbreviation when it is first introduced.
- Please correct the order of manuscript sections to: Abstract, Introduction, Results, Discussion, Materials and Methods, Acknowledgements, Disclosure and competing interests statement, References, Figure legends, Tables and their legends, Expanded View Figure legends.
- Materials and Methods:
 - o Mice experiments: please provide the origin of the mice, as well as gender, age, and housing and husbandry conditions.
 - o Cells: please provide the origin of the cells, and whether they were authenticated and tested for mycoplasma contamination.
 - o Antibodies: please provide dilutions/concentrations.
 - o Statistics: please provide a statement on sample size, exclusion/inclusion criteria, randomization, and blinding.
- The 'Data Availability' section should be placed after the Materials and Methods, and additionally contain database name and accession code.
- Funding should be merged with Acknowledgment. Please add the project number for ANR.
- Author contributions: CRediT has replaced the traditional author contributions section because it offers a systematic machine-readable author contributions format that allows for more effective research assessment. Please remove the Authors Contributions from the manuscript and use the free text boxes beneath each contributing author's name in our system to add specific details on the author's contribution. More information is available in our guide to authors.
- Replace "Conflicts of Interest" by "Disclosure statement and competing interests". We updated our journal's competing interests policy in January 2022 and request authors to consider both actual and perceived competing interests. Please review the policy <https://www.embopress.org/competing-interests> and update your competing interests if necessary.
- Please reformat the references with 10 author names (first names initials) listed before et al.; dois should be removed for all published works.

2/ Figures:

- Supplementary figures should be renamed "Figure EV1" etc. Expanded View (EV) Figures and Tables are collapsible/expandable online. Legends should be after main figure legends with the heading "Expanded View Figure Legends".
- Tables S1-3 should be renamed Table EV1 - Table EV3. Their legends need to be removed from the manuscript file and added to table files. Table files format should be updated to docx or .xlsx format.

- Figure legends:

1. Please note that a separate 'Data Information' section is required in the legends of figures 1c-d, e-f.
2. Please note that individual figure legends for supplementary figures 2a-b are not provided in the manuscript. This needs to be rectified.
3. Please indicate the statistical test used for data analysis in the legends of figures 2f; 3b, e-g; 4c; 5e; 8a-c, e; 9a-d, h, j, k, supplementary figures 1; 2a-b; 3b; 4b, d-e.
4. Please note that the box plots need to be defined in terms of minima, maxima, centre, bounds of box and whiskers, and percentile in the legends of figures 9c-d, h, j.
5. Please note that information related to n is missing in the legends of figures 2a; 5a-b, e; 6e; 8a-c; 9a-d, h, j, supplementary figures 1; 2a-b; 3a-b; 4b, d-e.
6. Please note that the error bars are not defined in the legends of figures 1d, f; 2b, d-e; 4c, f; 5a-e; 6a, c-d; 7b-c; 9a-b, e-f, k, supplementary figures 2a-b; 3a-b; 4b, d-e.
7. Please note that the scale bar needs to be defined for figures 2c; 4d-e.

3/ At EMBO Press we ask authors to provide source data for the main figures. Our source data coordinator will contact you to discuss which figure panels we would need source data for and will also provide you with helpful tips on how to upload and organize the files.

4/ Please provide a completed author checklist, which you can download from our author guidelines (<https://www.embopress.org/page/journal/17574684/authorguide#submissionofrevisions>). Please insert information in the checklist that is also reflected in the manuscript. The completed author checklist will also be part of the RPF.

5/ Our journal encourages inclusion of *data citations in the reference list* to directly cite datasets that were re-used and obtained from public databases. Data citations in the article text are distinct from normal bibliographical citations and should directly link to the database records from which the data can be accessed. In the main text, data citations are formatted as follows: "Data ref: Smith et al, 2001" or "Data ref: NCBI Sequence Read Archive PRJNA342805, 2017". In the Reference list, data citations must be labeled with "[DATASET]". A data reference must provide the database name, accession number/identifiers and a resolvable link to the landing page from which the data can be accessed at the end of the reference. Further instructions are available at .

6/ The paper explained: EMBO Molecular Medicine articles are accompanied by a summary of the articles to emphasize the major findings in the paper and their medical implications for the non-specialist reader. Please provide a draft summary of your article highlighting

7/ For more information: There is space at the end of each article to list relevant web links for further consultation by our readers. Could you identify some relevant ones and provide such information as well? Some examples are patient associations, relevant databases, OMIM/proteins/genes links, author's websites, etc...

8/ Every published paper now includes a 'Synopsis' to further enhance discoverability. Synopses are displayed on the journal webpage and are freely accessible to all readers. They include a short stand first (maximum of 300 characters, including space) as well as 2-5 one-sentences bullet points that summarizes the paper. Please write the bullet points to summarize the key NEW findings. They should be designed to be complementary to the abstract - i.e. not repeat the same text. We encourage inclusion of key acronyms and quantitative information (maximum of 30 words / bullet point). Please use the passive voice. Please attach these in a separate file or send them by email, we will incorporate them accordingly.

9/ As part of the EMBO Publications transparent editorial process initiative (see our Editorial at <http://embomolmed.embopress.org/content/2/9/329>), EMBO Molecular Medicine will publish online a Review Process File (RPF) to accompany accepted manuscripts.

This file will be published in conjunction with your paper and will include the anonymous referee reports, your point-by-point response and all pertinent correspondence relating to the manuscript. Let us know whether you agree with the publication of the RPF and as here, if you want to remove or not any figures from it prior to publication.

I look forward to receiving your revised manuscript.

Yours sincerely,

Lise Roth

Lise Roth, PhD

Senior Editor

EMBO Molecular Medicine

***** Reviewer's comments *****

Referee #1 (Remarks for Author):

The authors have adequately addressed my concerns. The in vivo data, in particular, supported the pathophysiological mechanisms described and their potential clinical impact.

Referee #2 (Remarks for Author):

The authors addressed my concerns successfully, including the in vivo confirmation of their major findings.

The authors addressed the minor editorial issues.

20th Mar 2024

Dear Dr. Chang-Marchand,

Thank you for submitting your revised files. Before I can accept your manuscript, please address the following editorial comments:

1/ Manuscript text:

- Materials and Methods/Statistics: please provide a statement on blinding, even if not blinding was done.
- Please check the information provided in the 'Data Availability' section as this section is restricted to datasets generated in this study.
- Your Table EV3 should be made a dataset (with legend in a separate tab).
- Figure legends:
 - o Please make sure all statistical tests used are mentioned in the figure legends (and for each panel).
 - o Please indicate both technical and biological replicates in figure 2a.
- Checklist: please complete the following:
 - o Author/manuscript information (top left corner)
 - o Experimental study design and statistics (every sub-section should be filled)
- The paper explained: I introduced minor modifications, please let me know if you agree or amend as you see fit, and include in the manuscript file:

Problem:

IFN γ is crucial for gut homeostasis and its dysregulation is linked to diverse colon pathologies, such as colitis and colorectal cancer (CRC). Drug resistance to chemotherapy limits the efficacy of standard CRC therapy. Given the important role of IFN γ /STAT1/PD-L1 axis in innate and adaptive immune responses, targeting the IFN γ signaling pathway could improve treatment efficacy for UC or CRC treatment.

Results:

CBX3 was identified as an antagonist of the IFN γ signaling cascade in the colon epithelium via repression of STAT1 and CD274 transcription. Upon IFN γ stimulation, CBX3 binding to STAT1 and CD274 promoters was decreased, concomitant with their increased gene expression. CBX3 depletion led to a strong increase of STAT1 and PD-L1 expression under IFN γ stimulation, suggesting a role for CBX3 in the control of IFN γ -stimulated immune gene activation. Accordingly, CBX3 deletion resulted in chronic mouse colon inflammation, accompanied by upregulated STAT1 and CD274 expression. In particular, CBX3 deletion heightened CRC cells' sensitivity to IFN γ , which ultimately enhanced their chemosensitivity both in vitro and in vivo.

Impact:

Our work identifies an interplay between CBX3 and IFN γ signaling that leads to regulation of the immune response in colon epithelium and of chemo-resistance of colorectal cancer. CBX3 appears as a potential target to enhance IFN γ -related therapeutic efficacy in UC and in colorectal cancer.

- Please resize your synopsis to 550 px wide x 300-600 px high. Please remove the information on BioRender from the image and add it to the Materials and Methods in a "Graphics" section.
- The synopsis text should have the following format:
 - o A short stand first (300 characters max, including space)
 - o 2-5 bullet points (30 words max)
- If you want to remove figures from your point-by-point letter, please let us know and we'll replace them by a standard "figure for referee removed" text. Please let us know if you agree with the publication of the Review Process File.

Looking forward to receiving your revised manuscript.

Yours sincerely,

Lise Roth

The authors addressed the remaining editorial issues.

3rd Apr 2024

Dear Dr. CHANG-MARCHAND,

Thank you for submitting your revised files. I am pleased to inform you that your manuscript is accepted for publication and is now being sent to our publisher to be included in the next available issue of EMBO Molecular Medicine!

Yours sincerely,

Lise Roth
